# MULTI-STUDENT DIFFUSION DISTILLATION FOR BETTER ONE-STEP GENERATORS

## ABSTRACT

Diffusion models achieve high-quality sample generation at the cost of a lengthy multistep inference procedure. To overcome this, diffusion distillation techniques produce student generators capable of matching or surpassing the teacher in a single step. However, the student model's inference speed is limited by the size of the teacher architecture, preventing real-time generation for computationally heavy applications. In this work, we introduce Multi-Student Distillation (MSD), a framework to distill a conditional teacher diffusion model into multiple single-step generators. Each student generator is responsible for a subset of the conditioning data, thereby obtaining higher generation quality for the same capacity. MSD trains multiple distilled students allowing smaller sizes and, therefore, faster inference. Also, MSD offers a lightweight quality boost over single-student distillation with the same architecture. We demonstrate MSD is effective by training multiple same-sized or smaller students on single-step distillation using distribution matching and adversarial distillation techniques. With smaller students, MSD gets competitive results with faster inference for single-step generation. Using 4 same-sized students, MSD sets a new state-of-the-art for one-step image generation: FID $1.20$ on ImageNet-$64\times64$ and $8.20$ on zero-shot COCO2014.

## 1 INTRODUCTION

Diffusion models are the dominant generative model in image, audio, video, 3D assets, protein design, and more (Ho et al., 2020; Kong et al., 2022; Blattmann et al., 2023; Anand and Achim, 2022; Nichol et al., 2022). They allow different conditioning inputs – such as class labels, text, or images – and achieve high-quality generated outputs. However, their inference process typically requires hundreds of model evaluations – with an often slow and bulky network – for a single sample. This procedure costs millions of dollars per day (Valyaeva, 2024; Google, 2024). It also prohibits applications requiring rapid synthesis, such as augmented reality. Real-time, low-cost, and high-quality generation will have huge financial and operational impacts while enabling new usage paradigms.

There has been a flurry of work on diffusion-distillation techniques to address the slow sampling of diffusion models (Luhman and Luhman, 2021; Song et al., 2023; Yin et al., 2024a). Inspired by knowledge distillation (Hinton et al., 2015), these methods use the trained diffusion model as a teacher and optimize a student model to match its generated output in as few as a single step. However, most diffusion distillation methods use the same student and teacher architecture. This prevents real-time generation for applications with bulky networks, such as video synthesis (Blattmann et al., 2023). While reducing model size can reduce inference time, it typically yields worse generation quality, thus presenting a speed-to-quality tradeoff dilemma with existing distillation methods.

We tackle this dilemma with our method, Multi-Student Distillation (MSD), that introduces multiple single-step student generators distilled from the pretrained teacher. Each student is responsible for a subset of conditioning inputs. We determine which student to use during inference and perform a single-model evaluation to generate a high-quality sample. This way, MSD enjoys the benefit of a mixture of experts (Jordan and Jacobs, 1994): it increases the model capacity without incurring more inference cost, thereby effectively pushing the limit of the speed-quality tradeoff.

When distilling into the same-sized students, MSD has the flexibility of being conceptually applicable to any distillation method for a performance boost. In addition, MSD allows distilling into multiple smaller students for reduced single-step generation time. Using smaller students prevents one from initializing a student from teacher weights, posing an additional technical challenge. We solve this challenge by adding a relatively lightweight score-matching pretraining stage before distillation (Sec. 4.3), and demonstrating its necessity and efficiency via extensive experiments.

We validate our approaches by applying MSD to distill the teacher into 4 same-sized students using distribution matching and adversarial distillation procedures (Yin et al., 2024a;b). The resulting students collectively outperform single-student counterparts, setting new state-of-the-art FID scores of 1.20 on one-step ImageNet-64×64 generation (Tab. 1) and 8.20 on one-step zero-shot COCO2014 generation (Tab. 2). We also distill the same teacher into 4 smaller students, which achieve a competitive FID of 2.88 on ImageNet (Tab. 1), with 42% less parameters per student.

We summarize our contributions below, which include:

- A new framework, MSD, that upgrades existing single-step diffusion distillation methods (Sec. 4.1) by increasing the effective model capacity without changing inference latency.
- Demonstrating the effectiveness of MSD by training multiple same sized students using SOTA distillation techniques (Yin et al., 2024a;b) in Sec. 4.2, resulting in new record FID scores in ImageNet-64×64 (Sec. 5.2) and zero-shot text-to-image generation (Sec. 5.3).
- A successful scheme to distill multiple smaller single-step students from the teacher model, achieving comparable generation quality with reduced inference time.

## 2 RELATED WORK

**Diffusion Sampling Acceleration.** While a line of work aims to accelerate diffusion models via fast numerical solvers for the PF-ODE (Lu et al., 2022a;b; Zheng et al., 2024; Karras et al., 2022; Liu et al., 2022), they usually still require more than 10 steps. Training-based methods that usually follow the knowledge distillation pipeline can achieve low-step or even one-step generation. Luhman and Luhman (2021) first used the diffusion model to generate a noise and image pair dataset that is then used to train a single-step generator. DSNO (Zheng et al., 2023) precomputes the denoising trajectory and uses neural operators to estimate the whole PF-ODE path. Progressive distillation (Salimans and Ho, 2022; Meng et al., 2023) iteratively halves the number of sampling steps required without needing an offline dataset. Rectified Flow (Liu et al., 2023a) and follow-up works (Liu et al., 2023b; Yan et al., 2024) straighten the denoising trajectories to allow sampling in fewer steps. Another approach uses self-consistent properties of denoising trajectories to inject additional regularization for distillation (Gu et al., 2023; Berthelot et al., 2023; Song et al., 2023; Song and Dhariwal, 2024; Luo et al., 2023; Ren et al., 2024; Kim et al., 2024).

The methods above require the student to follow the teacher's trajectories. Instead, a recent line of works aims to only match the distribution of the student and teacher output via variational score distillation (Yin et al., 2024a;b; Salimans et al., 2024; Xie et al., 2024a; Luo et al., 2024; Zhou et al., 2024a; Nguyen and Tran, 2024). The adversarial loss (Goodfellow et al., 2014), often combined with the above techniques, has been used to enhance the distillation performance further (Xiao et al., 2022; Zheng and Yang, 2024; Sauer et al., 2023a; 2024; Wang et al., 2023; Xu et al., 2024; Lin et al., 2024; Kim et al., 2024). Although MSD is conceptually compatible and offers a performance boost to all of these distillation methods, in this work, we demonstrate two specific techniques: distribution matching (Yin et al., 2024a) and adversarial distillation (Yin et al., 2024b).

**Mixture of experts training and distillation.** Mixture of Experts (MoE), first proposed in Jordan and Jacobs (1994), has found success in training very large-scale neural networks (Shazeer et al., 2017; Lepikhin et al., 2021; Fedus et al., 2022; Lewis et al., 2021; Borde et al., 2024). Distilling a teacher model into multiple students was explored by Hinton et al. (2015), and after that, has been further developed for supervised learning (Chen et al., 2020; Ni and Hu, 2023; Chang et al., 2022) and language modeling (Xie et al., 2024b; Kudugunta et al., 2021; Zuo et al., 2022). Although several works (Hoang et al., 2018; Park et al., 2018; Ahmetoğlu and Alpaydın, 2021) have proposed

MoE training schemes for generative adversarial networks, they train the MoE from scratch. This requires carefully tuning the multi-expert adversarial losses. eDiff-I (Balaji et al., 2022) uses different experts in different denoising timesteps for a multi-step diffusion model. A recent work (Zhou et al., 2024b) proposes to distill a pretrained diffusion model into an MoE for policy learning, which shares similar motivations with our work. However, to the best of our knowledge, MSD is the first method to *distill* multi-step teacher diffusion models into multiple one-step students for image generation.

**Efficient architectures for diffusion models.** In addition to reducing steps, an orthogonal approach aims to accelerate diffusion models with more efficient architectures. A series of works (Bao et al., 2022; Peebles and Xie, 2023; Hoogeboom et al., 2023) introduces vision transformers to diffusion blocks and trains the diffusion model with new architectures from scratch. Another line of work selectively removes or modifies certain components of a pretrained diffusion model and then either finetunes (Kim et al., 2023; Li et al., 2024; Zhang et al., 2024) or re-trains (Zhao et al., 2023) the lightweight diffusion model, from which step-distillation can be further applied (Li et al., 2024; Zhao et al., 2023). Our approach is orthogonal to these works in two regards: 1) In our method, each student only handles a subset of data, providing a gain in relative capacity. 2) Instead of obtaining a full diffusion model, our method employs a lightweight pretraining stage to obtain a good initialization for single-step distillation. Combining MSD with more efficient architectures is a promising future direction.

## 3 PRELIMINARY

We introduce the background on diffusion models in Sec. 3.1 and distribution matching distillation (DMD) in Sec. 3.2. We discuss how applying adversarial losses to improve distillation in Sec. 3.3.

### 3.1 DIFFUSION MODELS

Diffusion models learn to generate data by estimating the score functions (Song et al., 2021) of the corrupted data distribution on different noise levels. Specifically, at different timesteps $t$, the data distribution $p_{\text{real}}$ is corrupted with an independent Gaussian noise: $p_{t,\text{real}}(\boldsymbol{x}_t) = \int p_{\text{real}}(\boldsymbol{x}) q_t(\boldsymbol{x}_t|\boldsymbol{x}) d\boldsymbol{x}$ where $q_t(\boldsymbol{x}_t|\boldsymbol{x}) \sim \mathcal{N}(\alpha_t \boldsymbol{x}, \sigma_t^2 \boldsymbol{I})$ with predetermined $\alpha_t, \sigma_t$ following a forward diffusion process (Song et al., 2021; Ho et al., 2020). The neural network learns the score of corrupted data $\boldsymbol{s}_{\text{real}} := \nabla_{\boldsymbol{x}_t} \log p_{t,\text{real}}(\boldsymbol{x}_t) = -(\boldsymbol{x}_t - \alpha_t \boldsymbol{x})/\sigma_t^2$ by equivalently predicting the denoised $\boldsymbol{x}$: $\boldsymbol{\mu}(\boldsymbol{x}_t, t) \approx \boldsymbol{x}$. After training with the denoising score matching loss $\mathbb{E}_{\boldsymbol{x},t,\boldsymbol{x}_t}[\lambda_t \|\boldsymbol{\mu}(\boldsymbol{x}_t, t) - \boldsymbol{x}\|_2^2]$, where $\lambda_t$ is a weighting cofficient, the model generates the data by an iterative denoising process over a decreasing sequence of time steps.

### 3.2 DISTRIBUTION MATCHING DISTILLATION

Inspired by Wang et al. (2024), the works of Luo et al. (2024); Yin et al. (2024a); Ye and Liu (2024); Nguyen and Tran (2024) aim to train the single-step distilled student to match the generated distribution of the teacher diffusion model. This is done by minimizing the following reverse KL divergence between teacher and student output distributions, diffused at different noise levels for better support over the ambient space:

$$\mathbb{E}_t D_{\text{KL}}(p_{t,\text{fake}} \| p_{t,\text{real}}) = \mathbb{E}_{\boldsymbol{x}_t} \left( \log \left( \frac{p_{t,\text{fake}}(\boldsymbol{x}_t)}{p_{t,\text{real}}(\boldsymbol{x}_t)} \right) \right). \tag{1}$$

The training only requires the gradient of Eq. (1), which reads (with a custom weighting $w_t$):

$$\nabla_\theta \mathcal{L}_{\text{KL}}(\theta) := \nabla_\theta \mathbb{E}_t D_{\text{KL}} \simeq \mathbb{E}_{\boldsymbol{z},t,\boldsymbol{x}_t}[w_t \alpha_t (\boldsymbol{s}_{\text{fake}}(\boldsymbol{x}_t, t) - \boldsymbol{s}_{\text{real}}(\boldsymbol{x}_t, t)) \nabla_\theta G_\theta(\boldsymbol{z})], \tag{2}$$

where $\boldsymbol{z} \sim \mathcal{N}(0, \boldsymbol{I})$, $t \sim \text{Uniform}[T_{\min}, T_{\max}]$, and $\boldsymbol{x}_t \sim q(\boldsymbol{x}_t|\boldsymbol{x})$, the noise injected version of $\boldsymbol{x} = G_\theta(\boldsymbol{z})$ generated by the one-step student. Here, we assume the teacher denoising model accurately approximates the score of the real data, and a "fake" denoising model approximates the score of generated fake data:

$$\boldsymbol{s}_{\text{real}}(\boldsymbol{x}_t, t) \approx -\frac{\boldsymbol{x}_t - \alpha_t \boldsymbol{\mu}_{\text{teacher}}(\boldsymbol{x}_t, t)}{\sigma_t^2}, \quad \boldsymbol{s}_{\text{fake}}(\boldsymbol{x}_t, t) \approx -\frac{\boldsymbol{x}_t - \alpha_t \boldsymbol{\mu}_{\text{fake}}(\boldsymbol{x}_t, t)}{\sigma_t^2}. \tag{3}$$

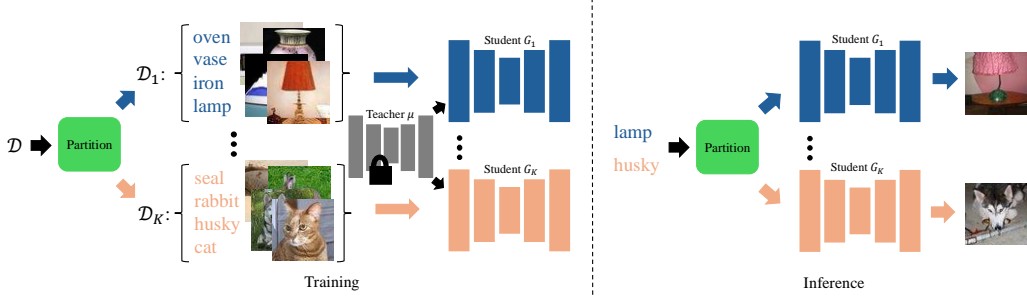

Figure 1: We visualize distilling into multiple students, where each student handles a subset of the input condition. At training, students are trained separately with filtered data. At inference, a single student is retrieved for generation given the corresponding input condition.

The "fake" denoising model is trained with the denoising objective with weighting $\lambda_t$:

$$\mathcal{L}_{\text{denoise}}(\phi) = \mathbb{E}_{\boldsymbol{z},t,\boldsymbol{x}_t}[\lambda_t \|\boldsymbol{\mu}_{\text{fake}}^{\phi}(\boldsymbol{x}_t, t) - \boldsymbol{x}\|_2^2]. \tag{4}$$

The generator and the "fake" denoising model are updated alternatively. To facilitate better convergence of the KL divergence, Distribution Matching Distillation (DMD) and DMD2 (Yin et al., 2024b) used two distinct strategies, both significantly improving the generation performance. DMD proposes to complement the KL loss with a regression loss to encourage mode covering:

$$\mathcal{L}_{\text{reg}}(\theta) = \mathbb{E}_{(\boldsymbol{z},y) \sim \mathcal{D}_{\text{paired}}} \ell(G_\theta(\boldsymbol{z}), y), \tag{5}$$

where $\mathcal{D}_{\text{paired}}$ is a dataset of latent-image pairs generated by the teacher model offline, and $\ell$ is the Learned Perceptual Image Patch Similarity (LPIPS) (Zhang et al., 2018). DMD2 instead applies a two-timescale update rule (TTUR), where they update the "fake" score model for $N$ steps per generator update, allowing more stable convergence. We use distribution matching (DM) to refer to all relevant techniques introduced in this section.

### 3.3 ENHANCING DISTILLATION QUALITY WITH ADVERSARIAL LOSS

The adversarial loss, originally proposed by Goodfellow et al. (2014), has shown a remarkable capability in diffusion distillation to enhance sharpness and realism in generated images, thus improving generation quality. Specifically, DMD2 (Yin et al., 2024b) proposes adding a minimal discriminator head to the bottleneck layer of the "fake" denoising model $\mu_{\text{fake}}$, which is naturally compatible with DMD's alternating training scheme and the TTUR. Moreover, they showed that one should first train the model without GAN to convergence, then add the GAN loss and continue training. This yields better terminal performance than training with the GAN loss from the beginning. We use adversarial distribution matching (ADM) to refer to distribution matching with added adversarial loss.

## 4 METHOD

In Sec. 4.1, we introduce the general Multi-Student Distillation (MSD) framework. In Sec. 4.2, we show how MSD is applied to distribution matching and adversarial distillation. In Sec. 4.3, we introduce an additional training stage enabling distilling into smaller students.

### 4.1 DISTILLING INTO MULTIPLE STUDENTS

We present Multi-Student Distillation (MSD), a general drop-in framework to be combined with any conditional single-step diffusion distillation method that enables a cheap upgrade of model capacity without impairing the inference speed. We first identify the key components of a single-step diffusion distillation framework and then present the modification of MSD.

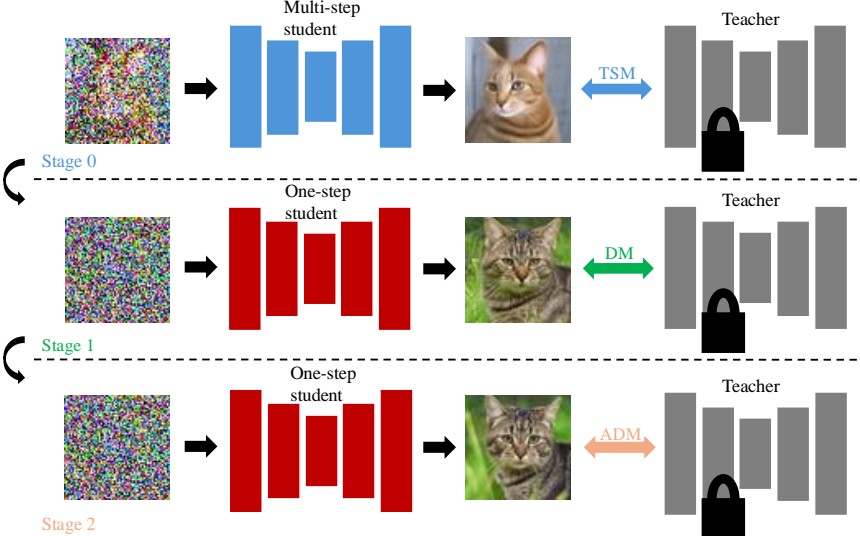

Figure 2: Three-stage training scheme in Eq. 9. Acronym meanings: TSM: teacher score matching (Eq. 8 & Eq. 9); DM: distribution matching (Eq. 9 & Sec. 3.2); ADM: adversarial distribution matching (Eq. 9 and Sec. 3.3). Stage 1 and Stage 2 are techniques from previous works that help with same-sized students; Stage 0 is our contribution, which is required for smaller students who cannot initialize with teacher weights.

In the vanilla one-student distillation, we have a pretrained teacher denoising diffusion model $\mu_{\text{teacher}}$, a training dataset $\mathcal{D}$, and a distillation method. The distillation yields a single-step generator $G(\boldsymbol{z}; y \in \mathcal{Y})$ via $G = \text{Distill}(\mu_{\text{teacher}}; \mathcal{D})$. The obtained generator $G$ maps a random latent $\boldsymbol{z}$ and an input condition $y$ into an image. In comparison, in an MSD scheme, we instead distill the teacher into $K$ different one-step generators $\{G_k(\boldsymbol{z}; y \in \mathcal{Y}_k)\}_{k=1}^K$ via

$$G_k = \text{Distill}(\mu_{\text{teacher}}, \mathcal{D}_k = F(\mathcal{D}, \mathcal{Y}_k)), \quad k = 1, ..., K \tag{6}$$

Specifically, each distilled student $G_k$ is specialized in handling a partitioned subset $\mathcal{Y}_k$ of the whole input condition set $\mathcal{Y}$. So, it is trained on a subset of the training data $\mathcal{D}_k \subset \mathcal{D}$, determined by $\mathcal{Y}_k$ via a filtering function $F$. Fig. 1 illustrates this idea.

The partition of $\mathcal{Y}$ into $\{\mathcal{Y}_k\}_{k=1}^K$ determines the input condition groups for which each student is responsible. As a starting point, we make the following three simplifications for choosing a partition:

- *Disjointness*: This prevents potential redundant training and redundant usage of model capacity.

- *Equal size*: Since students have the same architecture, the partitions $\{\mathcal{Y}_k\}_{k=1}^K$ should be of equal size that require similar model capacity.

- *Clustering*: Conditions within each partition should be more semantically similar than those in other partitions, so networks require less capacity to achieve a set quality on their partition.

The first two conditions can be easily satisfied in practice, while the third is not straightforward. For a class-conditional generation, partitioning by semantically similar and equal-sized classes serves a straightforward strategy, though extending it to text-conditional generation is nontrivial. Another promising strategy uses pretrained embedding layers such as the CLIP (Radford et al., 2021) embedding layer or the teacher embedding layer. One could find embeddings of the input conditions and then perform clustering on those embeddings, which are fixed-length numerical vectors containing implicit semantic information. We ablate partition strategies in Sec. 5.4.

The data filtering function $F$ determines the training subset data $\mathcal{D}_k$ from $\mathcal{Y}_k$. For example, a vanilla filtering strategy could set $F(\mathcal{D}, \mathcal{Y}_k) = \mathcal{D}_k := \mathcal{D}_{\mathcal{Y}_k}$, where $\mathcal{D}_{\mathcal{Y}_k}$ denotes the subset of the training dataset $\mathcal{D}$ that contains the desired condition $\mathcal{Y}_k$. Empirically, we found that this filtering works in most cases, although sometimes a different approach is justified, as demonstrated in Sec. 4.2.

## 4.2 MSD WITH DISTRIBUTION MATCHING

As a concrete example, we demonstrate the MSD framework using distribution matching (DM) and adversarial distillation techniques. Inspired by the two-stage framework in Yin et al. (2024b), each of our students is trained with a distribution matching scheme at the first stage and finetuned with an additional adversarial loss at the second stage (adversarial distribution matching, or ADM):

$$
\begin{aligned}
G_k^{(1)} &= \text{Distill}_{\text{DM}}\left(\boldsymbol{\mu}_{\text{teacher}}, F_{\text{DM}}(\mathcal{D}_{\text{DM}}, \mathcal{Y}_k)\right), \quad k = 1, ..., K, \\
G_k^{(2)} &= \text{Distill}_{\text{ADM}}\left(\boldsymbol{\mu}_{\text{teacher}}, F_{\text{ADM}}(\mathcal{D}_{\text{ADM}}, \mathcal{Y}_k); G_k^{(1)}\right), \quad k = 1, ..., K,
\end{aligned}
\tag{7}
$$

where we recall that $\boldsymbol{\mu}_{\text{teacher}}$ is the teacher diffusion model, $G_k^{(i)}$ is the $k$-th student generator at the $i$-th stage, $F$ is the data filtering function, $\mathcal{D}$ is the training data, and $\mathcal{Y}_k$ is the set of labels that student $k$ is responsible of. The first stage $\text{Distill}_{\text{DM}}$ uses distribution matching with either a complemented regression loss or the TTUR, with details in Sec. 3.2. These two methods achieve optimal training efficiency among other best-performing single-step distillation methods (Xie et al., 2024a; Zhou et al., 2024a; Kim et al., 2024) without an adversarial loss, with a detailed comparison in App. B.3. The second stage $\text{Distill}_{\text{ADM}}$ adds an additional adversarial loss (details in Sec. 3.3, which introduces minimal additional computational overhead and allows resuming from the first stage checkpoint, making it a natural choice.

**Designing the training data**  From Sec. 3, the data required for DM and ADM are $\mathcal{D}_{\text{DM}} = (\mathcal{D}_{\text{paired}}, \mathcal{C})$ and $\mathcal{D}_{\text{ADM}} = (\mathcal{D}_{\text{real}}, \mathcal{C})$, where $\mathcal{D}_{\text{paired}}, \mathcal{D}_{\text{real}}, \mathcal{C}$ represents generated paired data, real data and separate conditional input, respectively. We now discuss choices for the filtering function.

For the first stage data filtering $F_{\text{DM}}$, we propose $F_{\text{DM}}(\mathcal{D}_{\text{DM}}, \mathcal{Y}_k) = (\mathcal{D}_{\text{paired}}, \mathcal{C}_{\mathcal{Y}_k})$, where $\mathcal{C}_{\mathcal{Y}_k}$ denotes the subset of condition inputs $\mathcal{C}$ that contains $\mathcal{Y}_k$. In other words, we sample all input conditions only on the desired partition for the KL loss but use the whole paired dataset for the regression loss. This special filtering is based on the observation that the size of $\mathcal{D}_{\text{paired}}$ critically affects the terminal performance of DMD distillation: using fewer pairs causes mode collapse, whereas using more pairs challenge the model capacity. Naïvely filtering paired datasets by partition reduces the paired dataset size for each student and leads to worse performance, as in our ablation in App. B.2. Instead of generating more paired data to mitigate this imbalance, we simply reuse the original paired dataset for the regression loss. This is remarkably effective, which we hypothesize is because paired data from other input conditions provides effective gradient updates to the shared weights in the network.

For the second stage, we stick to the simple data filtering $F_{\text{ADM}}(\mathcal{D}_{\text{ADM}}, \mathcal{Y}_k) = (\mathcal{D}_{\text{real}, \mathcal{Y}_k}, \mathcal{C}_{\mathcal{Y}_k})$, so that both adversarial and KL losses focus on the corresponding partition, given that each student has enough mode coverage from the first stage.

## 4.3 DISTILLING SMALLER STUDENTS FROM SCRATCH

Via the frameworks presented in the last two sections, MSD enables a performance upgrade over alternatives for one student with the same model architecture. In this section, we investigate training multiple students with smaller architectures – and thus faster inference time – without impairing much performance. However, this requires distilling into a student with a different architecture, preventing initialization from pretrained teacher weights. Distilling a single-step student from scratch has previously been difficult (Xie et al., 2024a), and we could not obtain competitive results with the simple pipeline in Eq. 7. Therefore, we propose an additional pretraining phase $\text{Distill}_{\text{TSM}}$, with TSM denoting Teacher Score Matching, to find a good initialization for single-step distillation. TSM employs the following score-matching loss:

$$
\mathcal{L}_{\text{TSM}} = \mathbb{E}_t[\lambda_t \|\boldsymbol{\mu}_{\text{TSM}}^{\varphi}(\boldsymbol{x}_t, t) - \boldsymbol{\mu}_{\text{teacher}}(\boldsymbol{x}_t, t)\|_2^2],
\tag{8}
$$

where the smaller student with weights $\varphi$ is trained to match the teacher's score on real images at different noise levels. This step provides useful initialization weights for single-step distillation and

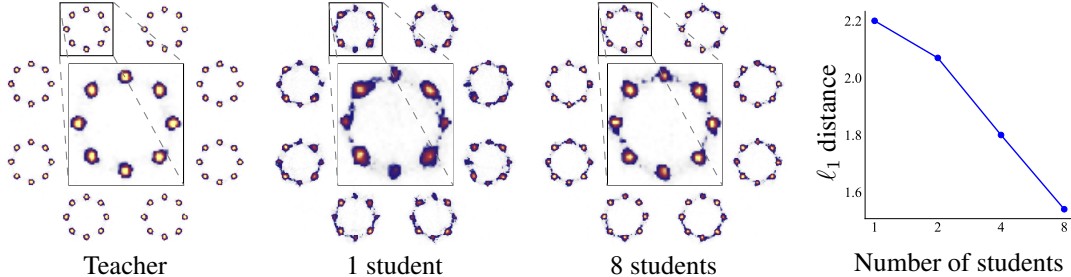

Figure 3: A 2D toy model. From left to right: teacher (multi-step) generation and student, one-step generation with $1$ and $8$ distilled students, the $\ell_1$ distance of generated samples between teacher and students. **Takeaway:** More students improve distillation quality on this easy-to-visualize setup.

is crucial to ensure convergence. With TSM added, the whole pipeline now becomes:

$$\boldsymbol{\mu}^{(0)} = \text{Distill}_{\text{TSM}}\left(\boldsymbol{\mu}_{\text{teacher}}, \mathcal{D}_{\text{real}}\right),$$

$$G_k^{(1)} = \text{Distill}_{\text{DM}}\left(\boldsymbol{\mu}_{\text{teacher}}, F_{\text{DM}}(\mathcal{D}_{\text{DM}}, \mathcal{Y}_k); \boldsymbol{\mu}^{(0)}\right), \quad k = 1, ..., K, \tag{9}$$

$$G_k^{(2)} = \text{Distill}_{\text{ADM}}\left(\boldsymbol{\mu}_{\text{teacher}}, F_{\text{ADM}}(\mathcal{D}_{\text{ADM}}, \mathcal{Y}_k); G_k^{(1)}\right), \quad k = 1, ..., K.$$

Although a smaller student may not perfectly match the teacher's score, it still provides a good initialization for stages 1 and 2. The performance gap is remedied in the latter stages by focusing on a smaller partition for each student. This three-stage training scheme is illustrated in Fig. 2.

## 5 EXPERIMENTS

To evaluate the effectiveness of our approach, we trained MSD with different design choices and compared against competing methods, including other single-step distillation methods.

In Sec. 5.1, we compare single vs multiple students on a 2D toy problem for direct visual comparison. For these experiments, we used only the DM stage. In Sec. 5.2, we investigate class-conditional image generation on ImageNet-64×64 (Deng et al., 2009) where we have naturally defined classes to partition. Here we explored training with the DM stage only, with both DM and ADM stages, and with all three stages for smaller students. We then evaluate MSD for a larger model in Sec. 5.3. We explored text-to-image generation on MS-COCO2014 (Lin et al., 2014) with varying training stages. We use the standard Fréchet Inception Distance (FID) (Heusel et al., 2017) score to measure generation quality. Comprehensive comparisons confirm that MSD outperforms single-student counterparts and achieves state-of-the-art performance in single-step diffusion distillation. Finally, in Sec. 5.4, we summarize our ablation experiments over design choices.

To focus on the performance boost from multi-student distillation, we applied minimal changes to the hyperparameters used by Yin et al. (2024a;b) for their distribution matching distillation implementations. More details on training and evaluation can be found in the App. D and E.

### 5.1 TOY EXPERIMENTS

In Fig. 3, we show the sample density of MSD with DM stage for a 2D toy experiment, where the real data distribution has $8$ classes, and each class is a mixture of $8$ Gaussians. We used a simple MLP with EDM schedules to train the teacher and then distill into $1$, $2$, $4$, and $8$ students for comparison. From the displayed samples and the $\ell_1$ distance from teacher generation, we observe that the collective generation quality increases as the number of students increases.

### 5.2 CLASS-CONDITIONAL IMAGE GENERATION

**Student architecture the same as the teacher:** We trained $K = 4$ students using the MSD framework and the EDM (Karras et al., 2022) teacher on class-conditional ImageNet-64×64 generation.

Table 1: Comparing class-conditional generators on ImageNet-64×64. The number of function evaluations (NFE) for MSD is 1 as a single student is used at inference for the given input.

| Method | NFE (↓) | FID (↓) |
|---|---|---|
| *Multiple Steps* | | |
| RIN (Jabri et al., 2023) | 1000 | 1.23 |
| ADM (Dhariwal and Nichol, 2021) | 250 | 2.07 |
| DPM Solver (Lu et al., 2022a) | 10 | 7.93 |
| Multistep CD (Heek et al., 2024) | 2 | 2.0 |
| *Single Step, w/o GAN* | | |
| PD (Salimans and Ho, 2022) | 1 | 15.39 |
| DSNO (Zheng et al., 2023) | 1 | 7.83 |
| Diff-Instruct (Luo et al., 2024) | 1 | 5.57 |
| iCT-deep (Song and Dhariwal, 2024) | 1 | 3.25 |
| Moment Matching (Salimans et al., 2024) | 1 | 3.0 |
| DMD (Yin et al., 2024a) | 1 | 2.62 |
| **MSD (ours): 4 students, DM only** | 1 | 2.37 |
| EMD (Xie et al., 2024a) | 1 | 2.20 |
| SiD (Zhou et al., 2024a) | 1 | 1.52 |
| *Single Step, w/ GAN* | | |
| Post-distillation, 4, 42% smaller students | 1 | 11.67 |
| **MSD (ours): 4, 42% smaller students, ADM** | 1 | 2.88 |
| StyleGAN-XL (Sauer et al., 2022) | 1 | 1.52 |
| CTM (Kim et al., 2024) | 1 | 1.92 |
| DMD2 (Yin et al., 2024b) | 1 | 1.28 |
| **MSD (ours): 4 students, ADM** | 1 | 1.20 |
| *teacher* | | |
| EDM (teacher, ODE) (Karras et al., 2022) | 511 | 2.32 |
| EDM (teacher, SDE) (Karras et al., 2022) | 511 | 1.36 |

Table 2: Comparing MSD to other methods on zero-shot text-to-image generation on MS-COCO2014. We measure speed with sampling time per prompt (latency) and quality with FID.

| Method | Latency (↓) | FID (↓) |
|---|---|---|
| *Unaccelerated* | | |
| DALL·E 2 (Ramesh et al., 2022) | - | 10.39 |
| LDM (Rombach et al., 2022) | 3.7s | 12.63 |
| eDiff-I (Balaji et al., 2022) | 32.0s | 6.95 |
| *GANs* | | |
| StyleGAN-T (Sauer et al., 2023b) | 0.10s | 12.90 |
| GigaGAN (Yu et al., 2022) | 0.13s | 9.09 |
| *Accelerated* | | |
| DPM++ (4 step) (Lu et al., 2022b) | 0.26s | 22.36 |
| InstaFlow-0.9B (Liu et al., 2023b) | 0.09s | 12.10 |
| UFOGen (Xu et al., 2024) | 0.09s | 12.78 |
| DMD (Yin et al., 2024a) | 0.09s | 11.49 |
| EMD (Xie et al., 2024a) | 0.09s | 9.66 |
| DMD2 (w/o GAN) | 0.09s | 9.28 |
| **MSD (ours): 4 students, DM only** | 0.09s | 8.80 |
| DMD2 (Yin et al., 2024b) | 0.09s | 8.35 |
| **MSD (ours): 4 students, ADM** | 0.09s | 8.20 |
| *teacher* | | |
| SDv1.5 (50 step, CFG=3, ODE) | 2.59s | 8.59 |
| SDv1.5 (200 step, CFG=2, SDE) | 10.25s | 7.21 |

We applied the simplest strategy for splitting classes among students: Each student is responsible for 250 consecutive classes in numerical order (i.e., $1/K$ of the 1000 classes). We compare the performance with previous methods and display the results in Tab. 1. Our DM stage, which uses the complementary regression loss, surpasses the one-student counterpart DMD (Yin et al., 2024a), achieving a modest drop of 0.25 in FID score, making it a strong competitor in single-step distillation without an adversarial loss. We then took the best pretrained checkpoints and trained with the ADM stage. The resulting model achieved the current state-of-the-art FID score of 1.20. It surpasses even the EDM teacher, StyleGAN-XL (Sauer et al., 2022), the multi-step RIN (Jabri et al., 2023) due to the adversarial loss. Fig. 4(a) and (b) display a comparison of sample generations, showing that our best students have comparable generation quality as the teacher.

**Student architecture smaller than the teacher:** Next, we trained 4 smaller student models with the prepended teacher score matching (TSM) stage from Sec. 4.3. This achieved a 42% reduction in model size and a 7% reduction in latency, with a slight degradation in FID score, offering a flexible framework to increase generation speed by reducing student size, and increasing generation quality by training more students. Fig. 4(c) displays sample generation from these smaller students, whereas Fig. 4(d) shows sample generations from an even smaller set of students, with a 71% percent reduction in model size and a 23% percent reduction in latency. We observed slightly degraded but still competitive generation qualities. Using more and larger students will further boost performance, as shown by ablations in Sec. 5.4 and App. B.4. Smaller students without the TSM stage fail to reach even proper convergence. Moreover, instead of the TSM stage, we performed post output distillation on best single-step checkpoints, and observed significant drop in performance. Hence the TSM stage is both necessary and efficient.

### 5.3 TEXT-TO-IMAGE GENERATION

**Student architecture the same as the teacher:** We evaluated the performance of text-to-image generation using the MS-COCO2014 (Lin et al., 2014) evaluation dataset. We distilled 4 students from Stable Diffusion (SD) v1.5 (Rombach et al., 2022) on a 5M-image subset of the COYO dataset (Byeon et al., 2022). For splitting prompts among students, we again employed a minimalist design: pass the prompts through the pre-trained SD v1.5 text encoder, pool the embeddings over the temporal dimension, and divide into 4 disjoint subsets along 4 quadrants. We trained with a classifier-free guidance (CFG) scale of 1.75 for best FID performance. Tab. 2 compares the evaluation results with previous methods. Our baseline method with only the DM stage again achieved a performance boost with a 0.48 drop in FID over the single-student counterpart DMD2 without adversarial loss

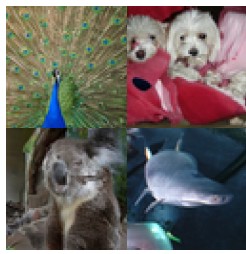 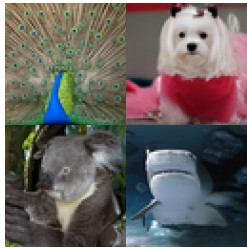 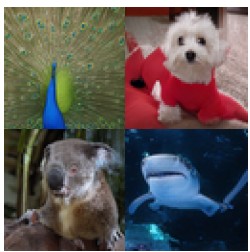 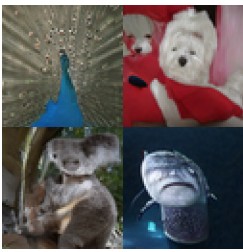

(a) Teacher (multistep)   (b) Same-sized students  (c) 42% smaller students (d) 71% smaller students

Figure 4: Sample generations on ImageNet-64×64 from the teacher and different sized students, with architecture and latency details in App. D. The same-size students have comparable or slightly better generation quality than the teacher. Smaller students achieve faster generation while still having decent qualities. Same-sized students are trained with DM and ADM stages, whereas smaller students are trained with all three stages (see Fig. 2).

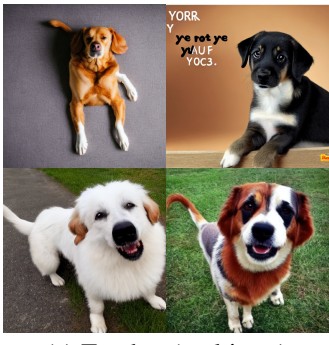 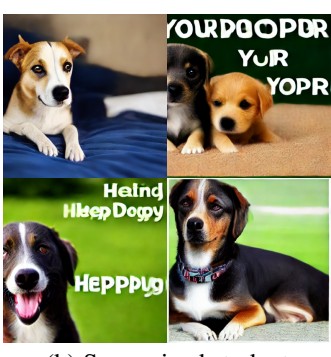 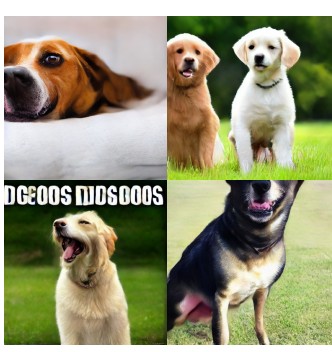

(a) Teacher (multistep)          (b) Same-sized student          (c) 83% smaller student

Figure 5: Samples on high guidance-scale text-to-image generations from the SD v1.5 teacher and different sized students, with full training details in App. D. The same-sized student has comparable quality to the teacher. The smaller student, trained on a subset of dog-related data, achieves faster generation while still having decent qualities. The same-sized student is trained with DM stage only, whereas the smaller student is trained with TSM and DM stages (see Fig. 2).

(Yin et al., 2024b). Continuing the ADM stage from the best checkpoint yielded a terminal FID of 8.20, again surpassing the single-student counterpart and achieving the current state-of-the-art FID score. In addition, for better visual quality, we also train with a larger CFG scale of 8, and display corresponding samples in Fig. 5(b) and App. F.2.

**Student architecture smaller than the teacher:** As a preliminary exploration, with the prepended teacher score matching (TSM) stage, we train a 83% smaller and 5% faster student on a dog-related prompt subset of COYO (containing $\sim 1\,210\,000$ prompts). We trained with a CFG scale of 8 and display the samples in Fig. 5. We observed fair generation quality despite a significant drop in model size. Improved training is likely to obtain better sample quality and generalization power. Due to limited computational resources and the complete coverage of the prompt set by the 4-student model, we did not train the full set of students at this size.

### 5.4 ABLATION STUDIES

Here, we ablate the effect of different components in MSD and offer insight into scaling. Unless otherwise mentioned, all experiments are conducted for class-conditional generation ImageNet-64×64, using only the DM stage for computational efficiency.

**MSD is still better with the same effective batch size.** To investigate if the performance boost from MSD comes from only a batch size increase over single student distillation, we make a comparison with the same effective batch size. As showcased in Tab. 3, MSD with 4 students and a batch size of 32 per student performs slightly better than the single-student counterpart with a batch size of 128, indicating that MSD likely benefits from a capacity increase than a batch size increase. As a takeaway, with a fixed training resource measured in processed data points, users are better off

distilling into multiple students with partitioned resources each than using all resources to distill into a single student. This is also reflected in our previous experiments, where we used significantly less resources per student than the single-student counterparts (see details in App. D). Although multiple students means multiple model weights to save, storage is often cheap, so in many applications, this cost is outweighed by our improved quality or latency.

**Simple splitting works surprisingly well.** We used consecutive splitting of classes in Sec. 5.2. Although it shows obvious advantage over random splitting, as shown in Tab. 3, it does not use the embedding information from the pretrained EDM model. Therefore, we investigated another strategy where we performed a $K$-means clustering ($K = 4$) on the label embeddings, resulting in 4 clusters of similar sizes: $(230, 283, 280, 207)$. However, MSD trained with these clustered partitions performs similarly to sequential partition, as shown in Tab. 3. For text-to-image generation, we performed $K$-means clustering on the pooled embeddings of prompts in the training data, re-

Table 3: Ablation studies on different components of MSD. All experiments are done on ImageNet-64$\times$64, trained with only the DM stage for 20k iterations, where $B$ is the batch size per student. See App. B.1.

| Method | FID ($\downarrow$) |
|---|---|
| 4 students, $B = 32$ | 2.53 |
| 1 students, $B = 128$ | 2.60 |
| 2 students, $B = 128$ | 2.49 |
| **4 students, B $= 128$ (baseline)** | 2.37 |
| 8 students, $B = 128$ | 2.32 |
| 4 students, $B = 128$, $K$-means splitting | 2.39 |
| 4 students, $B = 128$, random splitting | 2.45 |

sulting in clusters of vastly uneven sizes. Due to computational limitations, we opted for the simpler partition strategies outlined in Sec. 5.

**Effect of scaling the number of students.** In Tab. 3, we study the effect of increasing $K$, the number of distilled students. We kept the per-student batch size fixed so more students induce a larger effective batch size. We observe better FID scores for more students. We hypothesize that better training strategies, such as per-student tuning, will further improve the quality. Optimal strategies for scaling to ultra-large numbers of students is an interesting area for future work.

## 6 DISCUSSION

### 6.1 LIMITATIONS

MSD is the first work to explore diffusion distillation with multiple students, and it admits a few limitations that call for future work. 1) Further explorations could offer more insights into optimal design choices for a target quality and latency on various datasets, such as the number of students, input condition size for each student, and other hyperparameters. This is especially beneficial if the training budget is limited. 2) We apply simple partitioning for both class- and text-conditions and assign them disjointly to different students. Although our empirical study shows that simple alternatives do not offer obvious advantages, more sophisticated routing mechanisms may help. 3) We use simple channel reduction when designing smaller students to demonstrate feasibility. This results in a significantly smaller latency reduction than sample size reduction. Exploring other designs of smaller students will likely increase their quality and throughput. 4) We train different students separately, but we expect that carefully designed weight-sharing, loss-sharing, or other interaction schemes can further enhance training efficiency. 5) We hypothesize that MSD can be applied to other diffusion distillation methods and other modalities for similar benefits, but leave this for future work.

### 6.2 CONCLUSION

This work presented Multi-Student Distillation, a simple yet efficient method to increase the effective model capacity for single-step diffusion distillation. We applied MSD to the distribution matching and adversarial distillation methods. We demonstrated their superior performance over single-student counterparts in both class-conditional generation and text-to-image generation. Particularly, MSD with DMD2's the two-stage training achieves state-of-the-art FID scores. Moreover, we successfully distilled smaller students from scratch, demonstrating MSD's potential in further reducing the generation latency with multiple smaller student distillations. We envision building on MSD to enable generation in real-time, enabling many new use cases.

REPRODUCIBILITY

All implementation details are provided in App. D, and all evaluation details are provided in App. E.

ETHICS STATEMENTS

Our work aims to improve the quality and speed of diffusion models, thus we may inherit ethics concerns from diffusion models and generative models in general. Potential risks include fabricating facts or profiles that could mislead public opinion, displaying biased information that may amplify social biases, and displacing creative jobs from artists and designers.

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

Table 4: Glossary and notation

| | |
|---|---|
| MSD | Multi-student distillation |
| DM | Distribution Matching |
| ADM | Distribution Matching with Adversarial loss |
| TSM | Teacher Score Matching |
| MoE | Mixture of experts |
| DMD | Distribution Matching Distillation (Yin et al., 2024a) |
| DMD2 | Improved Distribution Matching Distillation (Yin et al., 2024b) |
| SOTA | State-of-the-art |
| FID | Fréchet Inception Distance |
| NFE | Number of Function Evaluations |
| SD | Stable Diffusion |
| TTUR | Two-Timescale Update Rule |
| CFG | Classifier-free guidance |
| MLP | Multi-layer Perceptron |
| GAN | Generative Adversarial Network |
| SDE/ODE | Stochastic/Ordinary Differential Equation |
| $i, j, k, n \in \mathbb{N}$ | Indices |
| $I, J, K, N \in \mathbb{N}$ | Sizes |
| $x, y, z \in \mathbb{R}$ | Scalars |
| $\boldsymbol{x}, \boldsymbol{y}, \boldsymbol{z} \in \mathbb{R}^N$ | Vectors |
| $\boldsymbol{X}, \boldsymbol{Y}, \boldsymbol{Z} \in \mathbb{R}^{N \times N}$ | Matrices |
| $\mathcal{X}, \mathcal{Y}, \mathcal{Z}$ | Sets / domains |
| $\boldsymbol{I}$ | The identity matrix |
| $G$ | Single-step generator |
| $\varphi$ | Student network weights |
| $\phi$ | "fake" score network weights |
| $\ell_1$ | Manhattan distance |
| Distill | Distillation method |
| $\mu$ | Denoising network |
| $K$ | Number of students |
| $k$ | Student index |
| $(i)$ | Distillation stage |
| $\mathcal{D}$ | Dataset |
| $\mathcal{C}$ | Condition dataset (without images) |
| $\mathcal{Y}$ | The abstract condition set |
| $F$ | Filtering function on input conditions |

## A ADDITIONAL EXPERIMENTAL RESULTS

### A.1 CLIP-SCORE FOR HIGH GUIDANCE SCALE

Table 5: CLIP-Score comparison for high guidance scale on MS-COCO2014. LCM-LoRA is trained with a guidance scale of 7.5, while all other methods use a guidance scale of 8.

| Method | Latency (↓) | CLIP-Score (↑) |
|---|---|---|
| DPM++ (4 step) Lu et al. (2022b) | 0.26s | 0.309 |
| UniPC (4 step) Zhao et al. (2024) | 0.26s | 0.308 |
| LCM-LoRA (1 step) Luo et al. (2023) | 0.09s | 0.238 |
| LCM-LoRA (4 step) Luo et al. (2023) | 0.19s | 0.297 |
| DMD2 (our reimplementation) Yin et al. (2024b) | 0.09s | 0.306 |
| **MSD4-ADM (ours)** | 0.09s | 0.308 |
| DMD Yin et al. (2024a) | 0.09s | 0.320 |
| SDv1.5 (teacher) Rombach et al. (2022) | 2.59s | 0.322 |

Tab. 5 shows the CLIP-Score of MSD and some single-student methods. MSD4-ADM achieves a competitive CLIP-Score, and beats the single student counterpart, DMD2. We believe the CLIP-Score can be further increased if one trains on the LAION dataset Schuhmann et al. (2022) instead of the COYO dataset Byeon et al. (2022).

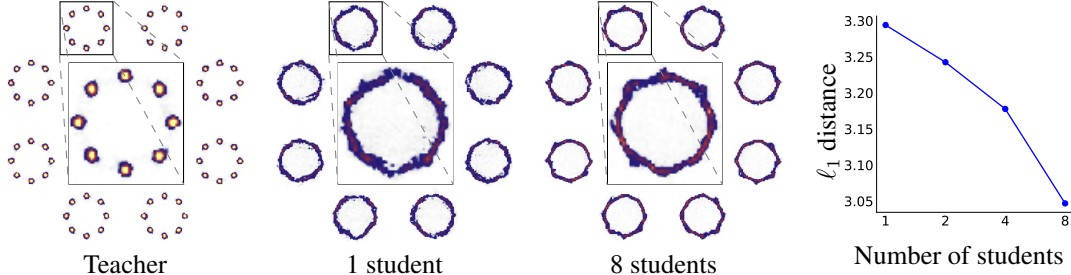

Figure 6: A 2D toy model, consistency distillation. From left to right: teacher (multi-step) generation and student, one-step generation with 1 and 8 distilled students, the $\ell_1$ distance of generated samples between teacher and students. **Takeaway:** More students improve distillation quality on this easy-to-visualize setup.

### A.2 CONSISTENCY DISTILLATION, TOY EXPERIMENTS

In order to show the wider applicability of MSD, we apply another distillation method, Consistency Distillation (Song et al., 2023), on the toy experiment setting in Sec. 5.1. Fig. 6 displays generated samples and the $\ell_1$ distance from teacher generation. While noting that consistency distillation achieves a weaker distillation of the teacher in general, we again observe better performance for more students. This indicates the generality of MSD.

## B ADDITIONAL ABLATION STUDIES

### B.1 TRAINING CURVES FOR SEC. 5.4

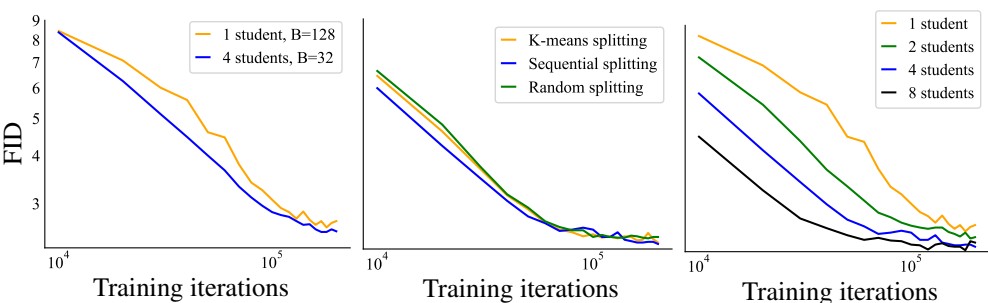

Figure 7: FID comparisons during training for ablations in Table 3.

Fig. 7 displays the training curves for the ablation studies shown in Tab. 3. The relative terminal performances are also reflected in the training process.

### B.2 THE EFFECT OF PAIRED DATASET SIZE ON DMD

In Sec. 4.2, we mentioned the special filtering strategy for MSD at DM stage: instead of partitioning the paired dataset for corresponding classes, we choose to keep the same complete dataset for each student. Fig. 8 demonstrates that the alternative strategy discourages mode coverage and leads to a worse terminal performance.

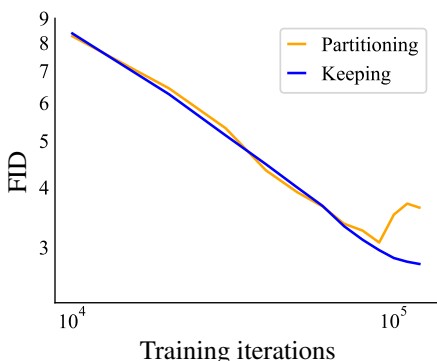

Figure 8: Comparison of paired dataset filtering strategies for MSD4-DM. Partitioning the paired dataset for each student discourages mode coverage, which results in worse terminal performance. In comparison, keeping the same paired dataset for each student achieves better performance without impairing the convergence speed.

### B.3 SINGLE-STEP DISTILLATION METHODS COMPARISON

Table 6: Comparison of various aspects of single-step distillation methods.

| Method on ImageNet | Terminal FID | Iterated Images |
|---|---|---|
| DMD (Yin et al., 2024a) (our reimplementation) | 2.54 | $\sim 130M$ |
| DMD2 (w/o GAN) (Yin et al., 2024b) | 2.61 | $\sim 110M$ |
| EMD (Xie et al., 2024a) | 2.20 | $\sim 600M$ |
| SiD ($\alpha = 1.0$) (Zhou et al., 2024a) | 2.02 | $\sim 500M$ |
| CTM (Kim et al., 2024) | >5 | $\sim 3M$ |

Table 6 justifies our choice of DMD/DMD2 as our first-stage training without adversarial loss. Competitor methods either need larger training data size (EMD, SiD), or have worse quality (CTM and other CM-based methods). DMD/DMD2, on the other hand, strike a good balance. We noticed DMD exhibits more stability for MSD on ImageNet, whereas DMD2 performs better for SD v1.5, which leads to our respective choices. As pointed out in App. D, we used a smaller-sized paired dataset (10 000 images) than the original DMD paper (25 000 images) for ImageNet, which significantly accelerated convergence without impairing the final performance. Moreover, as pointed out in Sec. 4.2, the same paired dataset can be used for all students, eliminating potential additional computation.

### B.4 MORE RESULTS ON DISTILLING INTO SMALLER STUDENTS

In Sec. 5.2, we trained MSD4-ADM on smaller students to demonstrate the tradeoff between generation quality and speed. Here, we make a more comprehensive ablation study on the interplay between student size, number of classes covered, and training stage, with results displayed in Fig. 9. We observe that generation quality increases with student size and decreases with more classes covered. MSD offers great flexibility for users to make these choices based on computational resources, generation quality, and inference speed requirements.

## C DEPLOYMENT SUGGESTIONS

Here we discuss some MSD deployment options for practitioners.

A naive option for deployment is to use increased GPU memory to host all models simultaneously. However, this is impractical and wasteful as only a single student model needs to be used for each user request. In settings with less GPU memory than all students' sum memory requirement, we must swap student models on and off GPUs. This incurs extra latency, however, in the few-GPU

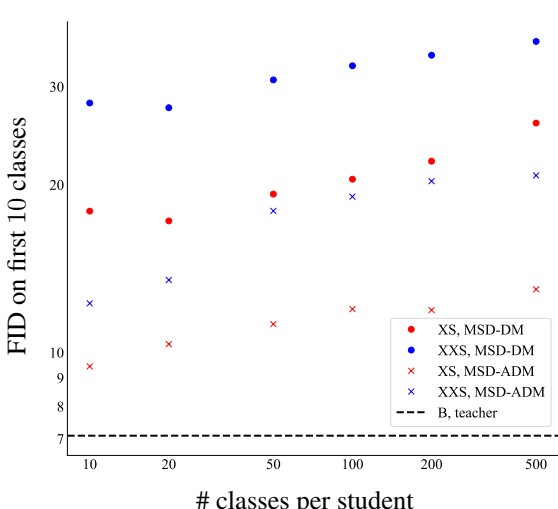

Figure 9: The effect of student size, number of classes covered, and training stage on the performance of a single smaller student. XS, XXS, and B refer to model sizes, with precise definitions provided in Tab. 7. Also, see special evaluation details in App. E.

many-users setting, there are already prominent latency issues, such as users needing to queue for usage. In few-user settings, resources are likely being taken offline to save cost and thus there is start-up latency for fresh requests too. Therefore, we argue that the more interesting setting is in large distributed deployment.

For settings with more GPU memory than all students' sum memory requirements, we can distribute the student models among a cluster of GPUs (as one would the teacher) and route each generation request to the appropriate student node. The routing layer is lightweight compared to the inference cost, so we pay little for it.

If the data has been partitioned uniformly according to user demand then the incoming requests are distributed uniformly among the student nodes. Therefore, we achieve equal throughput compared to the teacher without more overall model storage. However, finding such a partition is challenging, and user demand may change over time. This leaves finding the optimal allocation of resources to the student nodes an open problem. In practice, we expect that a reduced student model size would lead to an overall reduction in storage requirements compared to the teacher alone.

# D    IMPLEMENTATION DETAILS

## D.1    TOY EXPERIMENTS

The real dataset is a mixture of Gaussians. The radius for the outer circle is 0.5, the radius for the 8 inner circles is 0.1, and the standard deviation for each Gaussian (smallest circle) is 0.005. The teacher is trained with EDM noise schedule, where use $(\sigma_{\min}, \sigma_{\max}) = (0.002, 80)$ and discretized the noise schedule into 1000 steps. We train the teacher for $100\,000$ iterations with AdamW (Loshchilov, 2017) optimizer, setting the learning rate at 1e-4, weight decay to 0.01, and beta parameters to (0.9, 0.999). For distillation, we first generated dataset of 1000 pairs, then used DMD (Yin et al., 2024a) to train 1, 2, 4, and 8 students, respectively, all for $200\,000$ iterations, with reduced learning rate at 1e-7. We only sample the first 750 of the 1000 steps for distillation.

Each subfigure of Fig. 3 is a histogram with 200 bins on $100\,000$ generated samples, using a custom colormap. The loss is the mean absolute difference of binned histogram values.

## D.2    IMAGENET

### D.2.1    SAME-SIZED STUDENTS

Our ImageNet experiments setup closely follows DMD (Yin et al., 2024a) and DMD2 (Yin et al., 2024b) papers. We distill our one-step generators using the EDM (Karras et al., 2022) checkpoint "edm-imagenet-64x64-cond-adm". We use $\sigma_{\min} = 0.002$ and $\sigma_{\max} = 80$ and discretize the noise schedules into 1000 bins. The weight $w_t$ in Eq. (2) is set to $\frac{\sigma_t^2}{\alpha_t} \frac{CS}{\|\boldsymbol{\mu}_{\text{teacher}}(\boldsymbol{x}_t, t) - \boldsymbol{x}\|_1}$ where $S$ is the number of spatial locations and $C$ is the number of channels, and the weight $\lambda_t$ in Eq. (4) is set to $(\sigma_t^2 + 0.5^2)/(\sigma_t \cdot 0.5)^2$.

For the DM stage, we prepare a distillation dataset by generating $10\,000$ noise-image pairs using the deterministic Heun sampler (with $S_{\text{churn}} = 0$ over 256 steps. We use the AdamW optimizer (Loshchilov, 2017) with learning rate 2e-6, weight decay 0.01, and beta parameters (0.9,0.999). We compute the LPIPS loss using a VGG backbone from the LPIPS library (Zhang et al., 2018), and we upscale the image to $224 \times 224$ using bilinear upsampling. The regression loss weight is set to 0.25. We use mixed-precision training and a gradient clipping with an $\ell_2$ norm of 10. We partition the 1000 total classes into consecutive blocks of 250 classes and trained 4 specialized students using Distill$_{\text{DM}}$ and $F_{\text{DM}}$ defined in Sec. 7. Each student is trained on 4 A100 GPUs, with a total batch size of 128, for $200\,000$ iterations. This yields the MSD4-DM checkpoint in Tab. 1.

For the ADM stage, we attach a prediction head to the middle block of the fakescore model. The prediction head consists of a stack of $4 \times 4$ convolutions with a stride of 2, group normalization, and SiLU activations. All feature maps are downsampled to $4 \times 4$ resolution, followed by a single convolutional layer with a kernel size and stride of 4. The final output linear layer maps the given vector to a scalar predicted probability. We load the best generator checkpoint from DM stage, but re-initialize the fakescore and GAN classifier model from teacher weights, as we observed this leads to slightly better performance. We set the GAN generator loss weight to 3e-3 and the GAN discriminator loss weight to 1e-2, and reduce the learning rate to 5e-7. Each student is trained on 4 A100 GPUs, with a total batch size of 192, for $150\,000$ iterations. This yields the MSD4-ADM checkpoint in Tab. 1.

### D.2.2    SMALLER STUDENTS

Following a minimalist design, we pick our smaller student's architecture by changing hyperparameter values of the "edm-imagenet-64x64-cond-adm" checkpoint architecture. See details in Tab. 7.

For the MSD4-ADM-S checkpoint in Tab. 1, we train the TSM stage using the model architecture S with the continuous EDM noise schedule with $(P_{\text{mean}}, P_{\text{std}}) = (-1.2, 1.2)$ and the weighting $\lambda_t = (\sigma_t^2 + 0.5^2)/(\sigma_t \cdot 0.5)^2$. We use a learning rate of 1e-4. Each student is trained on 4 A100

Table 7: Hyperparameter details for different sized student models of the ADM architecture. Unspecified hyperparameters remain the same as the teacher. Latency is measured on a single NVIDIA RTX 4090 GPU.

| Model identifier | # channels | Channel multipliers | # Residual blocks | # parameters | latency |
|---|---|---|---|---|---|
| B (teacher) | 192 | [1,2,3,4] | 3 | 296M | 0.0271s |
| S | 160 | [1,2,2,4] | 3 | 173M | 0.0253s |
| XS | 128 | [1,2,2,4] | 2 | 86M | 0.0209s |
| XXS | 96 | [1,2,2,2] | 2 | 26M | 0.0192s |

GPUs, with a total batch size of 576, for $400\,000$ iterations. Then DM and ADM stages were trained using a total batch size of 160, following otherwise the same setup as Sec. D.2.1.

For the ablation study in B.4, we instead train a common TSM stage for all students for computational efficiency. We train this common stage using 16 A100 GPUs, with a total batch size of 3584 and 4864 for architecture XS and XXS, respectively. The DM and ADM stages are followed by specialized students with filtered data and 4 A100 GPUs each, with a total batch size of 224 and 256 for architecture XS and XXS, respectively, and using the same learning rate of 2e-6.

## D.3  SD v1.5

### D.3.1  SAME-SIZED STUDENTS, CFG=1.75

Our SD v1.5 experiments setup closely follows DMD2 (Yin et al., 2024b) paper. We distill our one-step generators from the SD v1.5 (Rombach et al., 2022) model, using a classifier-free guidance scale of $1.75$ for the teacher model to obtain the best FID score. We use the first 5M prompts from the COYO dataset (Byeon et al., 2022), and the corresponding 5M images for the GAN discriminator. We apply the DDIM noise schedule with 1000 steps for sampling $t$. The weight $w_t$ in Eq. 2 is set to $\frac{\sigma_t^2}{\alpha_t} \frac{CS}{\|\boldsymbol{\mu}_{\text{teacher}}(\boldsymbol{x}_t,t)-\boldsymbol{x}\|_1}$ where $S$ is the number of spatial locations and $C$ is the number of channels, and the weight $\lambda_t$ in Eq. 4 is set to $\alpha_t^2/\sigma_t^2$.

For the DM stage, we use the AdamW optimizer (Loshchilov, 2017) with learning rate 1e-5, weight decay 0.01, and beta parameters (0.9,0.999). We use gradient checkpointing, mixed-precision training, and a gradient clipping with an $\ell_2$ norm of 10. We partition the prompts and corresponding images by the 4 quadrants formed by the first 2 entries of the embeddings, where the embeddings are pooled from the outputs of the SD v1.5 text embedding layers. We choose not to include a regression loss but instead use a TTUR, which updates the fakescore model 10 times per generator update. Each of the 4 students is trained on 32 A100 GPUs, with a total batch size of 1536, for $40\,000$ iterations. This yields the MSD4-DM checkpoint in Tab. 2.

For the ADM stage, we attach a prediction head that has the same architecture (though with a different input size) as Sec. D.2. We load both the best generator checkpoint and the corresponding fakescore checkpoint from DM stage. We set the GAN generator loss weight to 1e-3 and the GAN discriminator loss weight to 1e-2, and reduce the learning rate to 5e-7. Each student is trained on 32 A100 GPUs, with a total batch size of 1024, for 5000 iterations. This yields the MSD4-ADM checkpoint in Tab. 2.

### D.3.2  SAME-SIZED STUDENTS, CFG=8

The above CFG=1.75 setting yields sub-optimal image quality. Similar to previous works (Yin et al., 2024a; Lin et al., 2024; Rombach et al., 2022), we choose CFG=8 for enhanced image quality. Due to time and computational resource limitations, we only train with the ADM stage. Each of the 4 students is trained on 32 A100 GPUs, with a learning rate of 1e-3 and a batch size of 1024 for both the fake and the real images, for 6000 iterations. This yields the checkpoint that is used to generate Fig. 5 (b) and Fig .12. Longer training with the added DM stage can likely further improve the generation quality.

### D.3.3 SMALLER STUDENT, CFG=8

We again pick our smaller student's architecture by changing the hyperparameter values of the SD v1.5 architecture. See details in Tab. 8.

Table 8: Hyperparameter details for different sized student models of the SD v1.5 architecture. Only the diffusion model part is measured since the text encoder and the VAE remain frozen. Unspecified hyperparameters remain the same as the teacher. Latency is measured on a single NVIDIA RTX 4090 GPU.

| Model identifier | # block_out_channels | # parameters | latency |
|---|---|---|---|
| B (teacher) | [320,640,1280,1280] | 860M | 0.041s |
| S | [160,320,320,640] | 142M | 0.039s |

To create a subset of dog-related data, we first selected $\sim 1\,210\,000$ prompts in the COYO Byeon et al. (2022) dataset whose embeddings are closest to "a dog." We then created an equal number of noise-image pairs from the SD v1.5 teacher using these prompts with CFG=8. We train the TSM stage using the model architecture S with the 1000-step DDIM noise schedule and the weighting $\lambda_t = \alpha_t^2/\sigma_t^2$. We use a learning rate of 1e-4. We then continue to the DM stage with the paired regression loss, using a learning rate of 1e-5, and finally continue to the ADM stage using generated paired images as "real" images with a learning rate of 5e-7. We use 16 A100 GPUs. The TSM stage is trained with a total batch size of 1536 for $240\,000$ iterations. The DM stage is trained with a total batch size of 512 for both the paired and the fake images for $20\,000$ iterations, and the ADM stage is trained with the same batch size for 6000 iterations. This yields the checkpoint used to generate Fig. 5(c). Longer training and better tuning are again likely to improve the generation quality further.

# E    EVALUATION DETAILS

For zero-shot COCO evaluation, we use the exact setup as GigaGAN (Yu et al., 2022) and DMD2 (Yin et al., 2024b). Specifically, we generate 30 000 images using the prompts provided by DMD2 code. We downsample the generated images using PIL to $256 \times 256$ Lanczos resizer. We then use the clean-fid (Parmar et al., 2022) to compute the FID score between generated images and 40 504 real images from the COCO 2014 validation dataset. Additionally, we use the OpenCLIP-G backbone to compute the CLIP score. For ImageNet, we generate 50 000 images and calculate FID using EDM (Karras et al., 2022) evaluation code. When selecting the best checkpoints for partitioned students, the same procedure is applied only for prompts/classes within respective partitions. For the ablation study in Sec. B.4, 10 000 images are generated for only the first 10 classes for an apple-to-apple comparison.

# F    ADDITIONAL QUALITATIVE RESULTS

## F.1    ADDITIONAL IMAGENET-$64 \times 64$ RESULTS

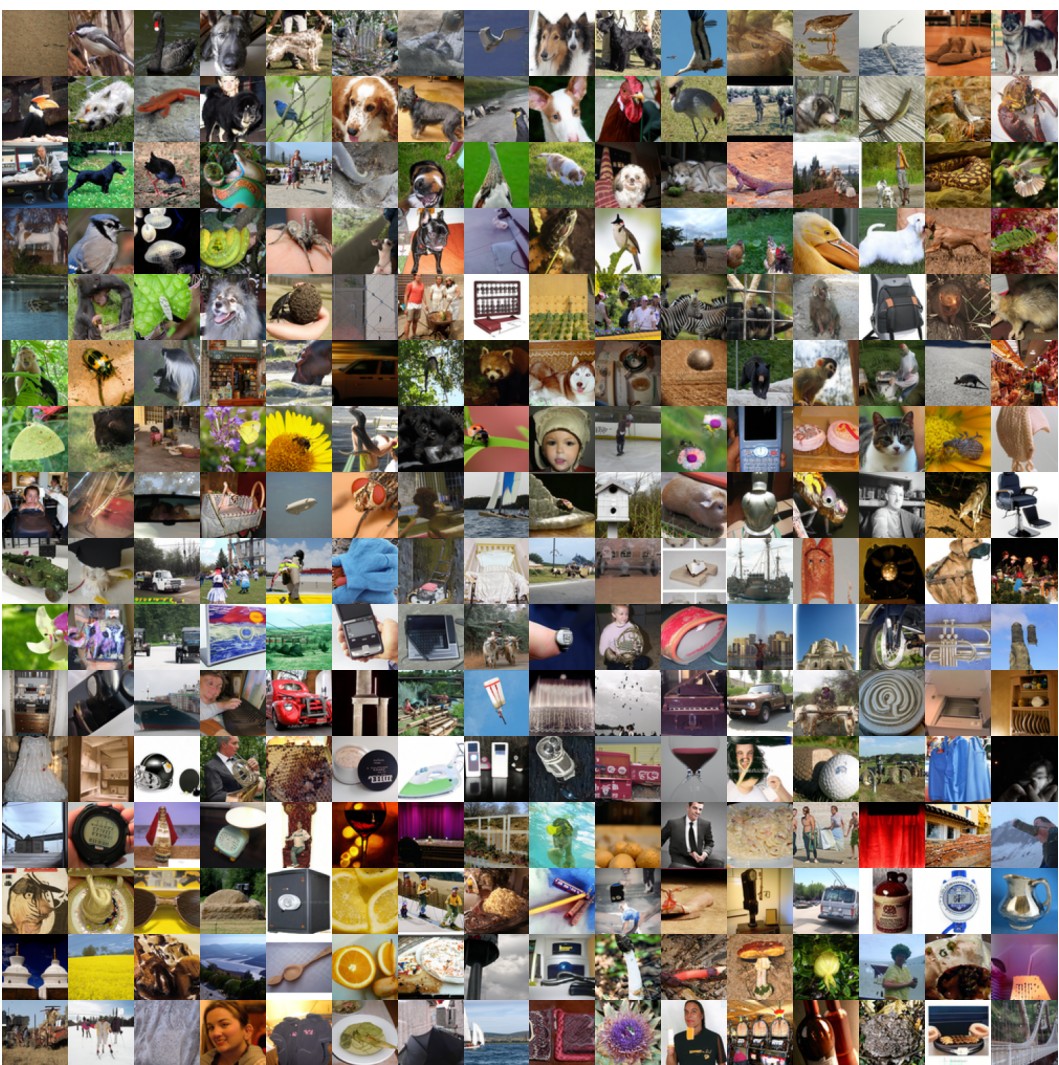

Figure 10: Collective one-step samples from 4 same-sized students trained with MSD-ADM on ImageNet (FID=1.20).

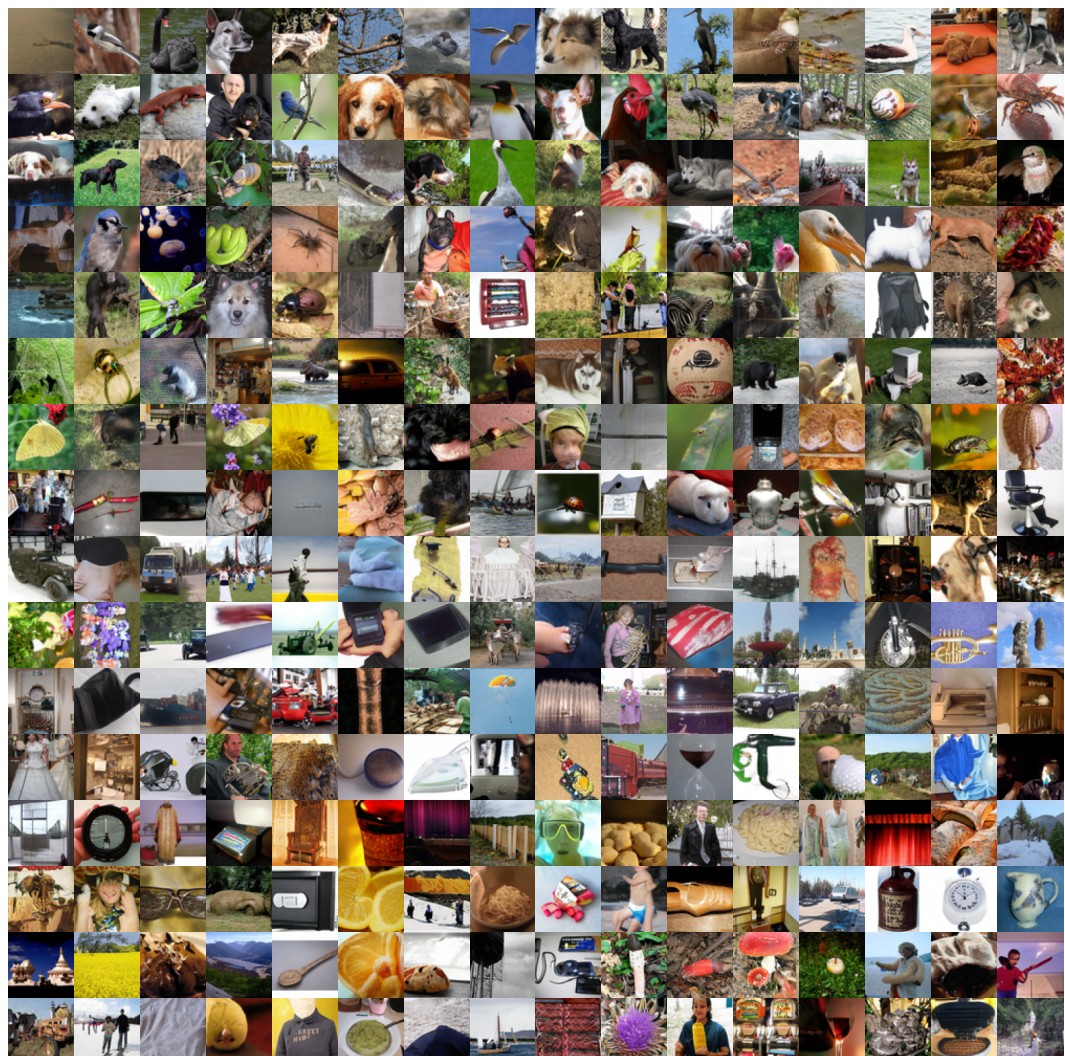

Figure 11: Collective one-step samples from 4 smaller students trained with MSD-ADM on ImageNet (FID=2.88).

In Fig. 10, we present more ImageNet-64×64 qualitative results collectively generated by our 4 same-sized students trained with MSD-ADM. In Fig. 11, we display corresponding generations from 4 smaller students with architecture S (see Tab. 7).

## F.2    ADDITIONAL TEXT-TO-IMAGE SYNTHESIS RESULTS

In Fig. 12, we present more text-to-image qualitative results collectively generated by our 4 students trained on the COYO dataset with MSD-ADM. These students are trained with a teacher classifier-free guidance (CFG) scale of 1.75, which yields sub-optimal visual qualities despite having a good FID score. Generation with better qualities can be obtained with a higher CFG scale.

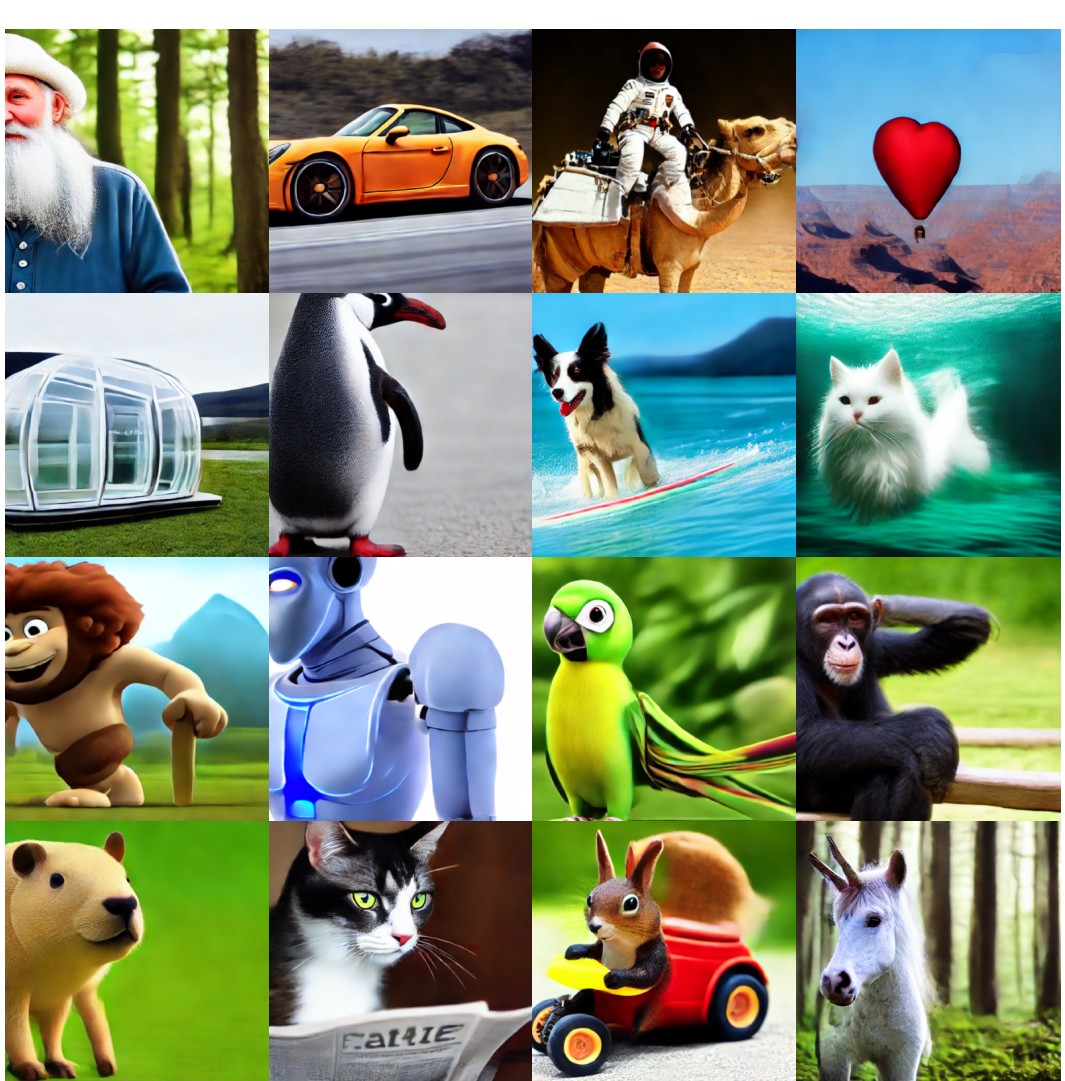

Figure 12: Collective one-step samples from 4 SD v1.5 students trained with MSD-ADM on COYO with CFG=8.

# G PROMPT DETAILS

## G.1 PROMPTS FOR FIG. 12

We use the following prompts for Fig. 12, from left to right, top to bottom:

- wise old man with a white beard in the enchanted and magical forest
- a high-resolution photo of an orange Porsche under sunshine
- Astronaut on a camel on mars
- a hot air balloon in shape of a heart. Grand Canyon
- transparent vacation pod at dramatic scottish lochside, concept prototype, ultra clear plastic material, editorial style photograph
- penguin standing on a sidewalk
- border collie surfing a small wave, with a mountain on background
- an underwater photo portrait of a beautiful fluffy white cat, hair floating. In a dynamic swimming pose. The sun rays filters through the water. High-angle shot. Shot on Fujifilm X
- 3D animation cinematic style young caveman kid, in its natural environment
- robot with human body form, robot pieces, knolling, top of view, ultra realistic
- 3D render baby parrot, Chibi, adorable big eyes. In a garden with butterflies, greenery, lush, whimsical and soft, magical, octane render, fairy dust
- a chimpanzee sitting on a wooden bench
- a capybara made of voxels sitting in a field
- a cat reading a newspaper
- a squirrell driving a toy car
- close-up photo of a unicorn in a forest, in a style of movie still

## G.2 PROMPTS FOR FIG. 5

We use the following prompts (same for all three models), from left to right, top to bottom:

- dog on a bed
- Your Puppy Your Dog
- Trained Happy Dog
- Very handsome dog.

