# OpenReview forum: "Multi-Student Diffusion Distillation for Better One-Step Generators"
_ICLR.cc/2025/Conference — Submitted to ICLR 2025_

### Official Review · Reviewer_WJfY · 2024-11-02

**Soundness:** 2
**Presentation:** 2
**Contribution:** 2
**Rating:** 3
**Confidence:** 3

**Summary:**

The paper introduces Multi-Student Distillation (MSD), an approach for diffusion model distillation, which improves existing single-step methods by increasing effective model capacity without added inference latency. MSD uses multiple student models, each optimized for a subset of conditioning inputs, to generate samples in a single step. This framework enhances flexibility by supporting multiple smaller student models to reduce generation time and enables initialization without requiring teacher weights. The authors validate MSD through experiments, achieving improved FID scores on various benchmarks with reduced parameters and comparable generation quality.

**Strengths:**

+ The organization and writing of the paper are clear, making it easy to understand and follow, and the review of related work is thorough.
+ The discussion on data partitioning is valuable and aligns well with the positioning of the proposed method. I also suggest the authors conduct more comprehensive experimental validation on this aspect.

**Weaknesses:**

- The technical novelty is limited, as the approach simply extends the distillation of a teacher diffusion model to multiple student diffusion models. The distillation methods used, DM and ASD, are existing techniques, so the improvement offered by this approach is marginal.
- The performance improvement is also marginal. As seen in Tables 1 and 2, the improvement of MSD over DMD2 is small, yet it requires more training and model resources. Although the paper explores smaller student models, Table 1 shows a significant drop in performance, which reduces the practical contribution of this work in terms of lowering inference costs.

**Questions:**

Does this method require more model storage for practical deployment? Do the authors have any solutions to address this issue?

---

> ### Author Response · Authors · 2024-11-20
> **Response to Reviewer WJfY**
>
> Thank you for the thoughtful and helpful feedback.
>
>     The technical novelty is limited, as the approach simply extends the distillation of a teacher diffusion model to multiple student diffusion models. The distillation methods used, DM and ASD, are existing techniques, so the improvement offered by this approach is marginal.
>
> We note that several technical challenges must be addressed to achieve multi-student distillation. Particularly when using smaller students initialized from scratch. Specifically, we introduced a teacher score-matching stage to provide a good initialization for the student. We performed a thorough ablation study to demonstrate its necessity and efficiency (see Table 1 and lines 418-421). Moreover, our method can be applied relatively seamlessly on top of existing distillation techniques. This means that we can benefit from additional advances in the distillation literature and provide a boost on top of any approach + methods to reduce student size without compromising capacity as much. Given that MSD strictly improves the performance of the baseline methods we implemented on top of (DMD & DMD2), we argue that these innovations will interest the research community and practitioners. We clarify these details further in the paper now.
>
>     The performance improvement is also marginal. As seen in Tables 1 and 2, the improvement of MSD over DMD2 is small, yet it requires more training and model resources. Although the paper explores smaller student models, Table 1 shows a significant drop in performance, which reduces the practical contribution of this work in terms of lowering inference costs.
>
> We argue that the boost to FID is not insignificant. We successfully provide improved performance over state-of-the-art approaches for multiple teacher models on several datasets. Furthermore, since submission, we have improved the quality of the small students (see the updated Figure 5). Note that the students are significantly smaller (<20% of the teacher size) — competing pruning methods typically provide a much smaller reduction in model size. While better pruning techniques are not the focus of this work, we believe they can further improve the performance of the smaller students.
>
>     Does this method require more model storage for practical deployment? Do the authors have any solutions to address this issue?
>
> Thank you for raising this question. This point is of general interest to practitioners and we have thus included a discussion in Appendix C that reads as follows.
>
> A naive option for deployment is to use increased GPU memory to host all models simultaneously. However, this is impractical and wasteful as only a single student model needs to be used for each user request. In settings with less GPU memory than all students’ sum memory requirement, we must swap student models on and off GPUs. This incurs extra latency, however, in the few-GPU many-users setting, there are already prominent latency issues, such as users needing to queue for usage. In few-user settings, resources are likely being taken offline to save cost and thus there is start-up latency for fresh requests too. Therefore, we argue that the more interesting setting is in large distributed deployment.
>
> For settings with more GPU memory than all students’ sum memory requirements, we can distribute the student models among a cluster of GPUs (as one would the teacher) and route each generation request to the appropriate student node. The routing layer is lightweight compared to the inference cost, so we pay little for it.
>
> If the data has been partitioned uniformly according to user demand then the incoming requests are distributed uniformly among the student nodes. Therefore, we achieve equal throughput compared to the teacher without more overall model storage. However, finding such a partition is challenging, and user demand may change over time. This leaves finding the optimal allocation of resources to the student nodes an open problem. In practice, we expect that a reduced student model size would lead to an overall reduction in storage requirements compared to the teacher alone.

---

> ### Author Response · Authors · 2024-11-25
> **Request for further discussion**
>
> Hello,
>
> Thank you for taking the time to provide a careful review of our submission. As the discussion period is nearing an end, we hope that you will evaluate our revisions and response. We believe that we have addressed all points of concern and clarification in your original review and would greatly appreciate the opportunity to discuss these with you further.
>
> Thank you.

---

### Official Review · Reviewer_D8t3 · 2024-11-03

**Soundness:** 2
**Presentation:** 2
**Contribution:** 2
**Rating:** 3
**Confidence:** 4

**Summary:**

This paper addresses the high computational cost associated with multistep inference in diffusion models by focusing on the speed-quality tradeoff in distillation. The authors propose a Multi-Student Distillation (MSD) framework to enhance both generation speed and output quality. In this framework, a teacher model is distilled into several single-step student models, each specialized for generating data under specific input conditions.

**Strengths:**

Overall, this paper is well-written and easy to follow, with relatively new comparison methods.

**Weaknesses:**

1. The authors state in Line 257 that 'Conditions within each partition should be more semantically similar than those in other partitions, so networks require less capacity to achieve a set quality on their partition.' However, there are no experiments presented to support this claim. I believe that implementing this idea is challenging and will demand additional computational resources. I recommend including relevant experiments and source code to facilitate a comprehensive review.

2. The statement in Line 15 that 'the student model’s inference speed is limited by the size of the teacher architecture' is misleading, as the inference speed of the student model is independent of the teacher model; the student only depends on the teacher during the distillation training phase. I recommend proofreading the entire paper to ensure clarity and professionalism.

3. The proposed method introduces multiple student models; therefore, comparisons and analyses of the model parameters should be a focal point of the paper.

4. The proposed method leverages adversarial distillation with the expectation of enhancing the distillation effect. However, there is no comparison with standard distillation methods or other variants to validate the adversarial distillation’s anticipated advantages.

5. In the ablation studies section, only a quantitative analysis of the generation effect is presented. I believe that a qualitative analysis should also be included, as the paper aims to enhance generation quality.

6. I can not conduct a comprehensive review of the technological accuracy of this paper, as it is empirical rather than theoretical, and the implementation code is not provided.

**Questions:**

Please refer to the Weaknesses and Questions.

---

> ### Author Response · Authors · 2024-11-20
> **Response to Riewer D8t3**
>
> Thank you for the thoughtful and helpful feedback.
>
>     The authors state in Line 257 that 'Conditions within each partition should be more semantically similar than those in other partitions, so networks require less capacity to achieve a set quality on their partition.' However, there are no experiments presented to support this claim. I believe that implementing this idea is challenging and will demand additional computational resources.
>
> This line refers to a general guideline for partitioning the input data. We achieve this by sequentially partitioning semantically ordered ImageNet classes (Section 5.2), label clustering in our ablations (Table 3), and clustering text prompts for SD experiments (Section 5.3). We disagree that implementing this is challenging as this structure exists in many datasets of interest, and we have demonstrated several ways to achieve this.
>
>     The statement in Line 15 that 'the student model’s inference speed is limited by the size of the teacher architecture' is misleading, as the inference speed of the student model is independent of the teacher model; the student only depends on the teacher during the distillation training phase. I recommend proofreading the entire paper to ensure clarity and professionalism.
>
> In the vast majority of diffusion distillation works, the student architecture matches the teacher architecture [1,2,3,4]. In fact, without this assumption, diffusion distillation is significantly harder to achieve. We were referring to this when we said the teacher architecture constrains the inference speed.
>
> [1] Song, Yang, et al. "Consistency Models." International Conference on Machine Learning. PMLR, 2023.
>
> [2] Yin, Tianwei, et al. "One-step diffusion with distribution matching distillation." Proceedings of the IEEE/CVF Conference on Computer Vision and Pattern Recognition. 2024.
>
> [3] Xie, Sirui, et al. "EM Distillation for One-step Diffusion Models." arXiv preprint arXiv:2405.16852 (2024).
>
> [4] Liu, Xingchao, Chengyue Gong, and Qiang Liu. "Flow Straight and Fast: Learning to Generate and Transfer Data with Rectified Flow." The Eleventh International Conference on Learning Representations (ICLR). 2023.
>
>     The proposed method introduces multiple student models; therefore, comparisons and analyses of the model parameters should be a focal point of the paper.
>
> Please clarify what you mean by this, as we are unclear on what type of parameters or comparisons/analyses you intend.
>
> For our method, there is an increase in storage of model parameters, but this is vanishingly cheap compared to the compute requirements. Additional students can be trained separately and used separately at inference. This does not require more memory nor complicated orchestration — we simply choose the student at inference time depending on the prompt.
>
> We have also added additional details to the paper clarifying our method’s storage usage.
>
> We intentionally kept the same set of hyperparameters to illustrate the simplicity of our framework.
>
> If you provide specific examples of comparisons or analyses you’d like to see, we can try including them.
>
>     The proposed method leverages adversarial distillation with the expectation of enhancing the distillation effect. However, there is no comparison with standard distillation methods or other variants to validate the adversarial distillation’s anticipated advantages.
>
> We include results with and without adversarial objectives. See Tables 1 and 2 and Figure 8 for more detailed comparisons.
>
>     In the ablation studies section, only a quantitative analysis of the generation effect is presented. I believe that a qualitative analysis should also be included, as the paper aims to enhance generation quality.
>
> We include several figures showing samples generated from our proposed method and those of the teacher and baseline methods (e.g., Figures 4 & 5). If you feel something specific is missing, please advise us, and we would happily include additional qualitative analysis.
>
> We greatly appreciate your attention and hope our response adequately addresses your concerns. Please let us know if you have any unresolved issues, and thanks for your help in this process.

---

> > ### Comment · Reviewer_D8t3 · 2024-11-25
> > **Confusion about the Response**
> >
> > > This line refers to a general guideline for partitioning the input data. We achieve this by sequentially partitioning semantically ordered ImageNet classes (Section 5.2), label clustering in our ablations (Table 3), and clustering text prompts for SD experiments (Section 5.3). We disagree that implementing this is challenging as this structure exists in many datasets of interest, and we have demonstrated several ways to achieve this.
> >
> > Thank you for your response. However, I feel that it did not fully address my question clearly and constructively. To confirm my understanding, Table 3 presents the ablation study on the number of students, and the ablation experiment on batch size is correct, correct? Additionally, Tables 1 and 2 focus on ablation studies for the student parameter count as well as teacher score matching, distribution matching, and adversarial distribution matching. Is that right?
> > I still do not see an ablation experiment that directly addresses my original question—whether clustering methods were used to classify different categories. Could you clarify how your response relates to this specific aspect? I would appreciate further clarification to better understand your approach to my concern. Thank you!
> >
> >
> >
> > I believe your article as a whole lacks significant innovation, but there is one innovative idea you mentioned that caught my attention, which is on line 257 of the paper. You stated that the division of **y labels** determines the input condition group that each student model is responsible for. You also mentioned that the conditions within each partition should be semantically more similar than those in other partitions, and clustering methods were used to ensure that the content generated by each student model is semantically more coherent. However, your subsequent experiments did not provide further confirmation of the effectiveness of this approach.
> >
> > What I suggest is that you could directly compare the method of **not using clustering** (and not ensuring that the student model generates semantically similar content) with the method of **using clustering** (and ensuring semantic similarity in the content generated by the student model). Specifically, you could compare the FID (Fréchet Inception Distance) values of the images generated by the model under these two different training settings. By demonstrating and analyzing the generated results, you would be able to verify the effectiveness of the clustering methods. Additionally, this would quantitatively validate how much improvement the clustering approach brings to the model's performance. This comparison could significantly strengthen your argument and substantiate the innovative value of the clustering approach.
> >
> > Based on the provided response, I do not feel that my concerns were fully addressed. As such, I prefer to keep my score unchanged. Thank you for your efforts.

---

> ### Author Response · Authors · 2024-11-25
> **Response to Reviewer D8t3**
>
> Thanks for your response.
>
> We detailed in Table 3 as well as line 492-505 that we performed an ablation study on clustering for ImageNet64: we compared three different approaches: 1) random splitting 2) sequential splitting 3) splitting by **K-means clustering**. The numbers in Table 3 suggest that sequential splitting and splitting by clustering have similar performance, both better than random splitting. This indicates that sequential splitting does ensure semantic similarity (this can be confirmed by looking at the numeric order of ImageNet classes). For SD1.5, we also explicitly mentioned how to perform **clustering** in line 427 and line 1167. Please let us know if you still find it confusing.
>
> We would also like to point out that a **significant innovation** of our approach is that we are the first to distill into a single-step student with **a smaller architecture** (i.e. the teacher score matching stage). We demonstrated that this stage is both **necessary** and **efficient** by conducting relevant ablation studies (see Table 1 and line 418-421).
>
> We greatly appreciate your attention and hope our response adequately addresses your concerns. Please let us know if you have any unresolved issues, and thanks for your help in this process.

---

### Official Review · Reviewer_igyA · 2024-11-06

**Soundness:** 4
**Presentation:** 3
**Contribution:** 3
**Rating:** 8
**Confidence:** 5

**Summary:**

This paper introduces a 'Multi-Student Diffusion Distillation' framework. The core idea behind the proposed method stems from Mixture-of-Experts. Particularly, the paper proposes to distill a pre-trained Diffusion model (Teacher model) into multiple  Student model, where each Student is responsible for learning of a subset of conditions. This effectively increases the model capacity by amortizing the set of conditions into smaller subsets where a smaller Student model is responsible for corresponding subset. There are few keypoints pertaining to the proposed method: (a) partitioning/filtering function to partition the set of conditions into subsets, some of the desired features of such function are described in Section 4.1; (b) distillation into multiple Student models, where each Student is responsible for a subset of conditioning variables; (c) support for smaller-sized Student model unlike previous methods which employ same-sized Student model; (d) a teacher score matching phase for smaller-sized Student networks for initialization and better training. The paper primarily deals with Distribution Matching Distillation (DMD) and its extension Adversarial Distribution Matching (ADM). The proposed method SoTA FID on ImageNet 64x64 for one-step generation. The paper is well written and presented. The idea is very intuitive, however, it is interesting to see it working in practice on models like StableDiffusion.

**Strengths:**

1. The paper is well written and presented. I enjoyed reading the paper. Though MoE is not a new idea, using it for Distillation is new, further using it to accelerate inference is commendable.
2. The idea of using Multiple-Students for distillation for inference time-quality tradeoff is quite intuitive. Moreover, assigning a student to a subset of conditions is a smart choice to increase the capacity of overall model.
3. Authors solve the obvious problem with above choice - initialization from scratch - by introducing an additional TSM stage which gives a good initialization, allowing for further distillation stage.
4. The empirical results are quite strong and encouraging. The proposed method achieves SoTA FID on Imagenet 64x64. Further, it shows encouraging results on distilling StableDiffusion performing better than several one-step generation methods.

**Weaknesses:**

1. The paper focuses exclusively on DMD (Distribution Matching Distillation) and its extension ADA, which limits the demonstration of the method's generality. While the authors acknowledge this limitation, can the authors demonstrate preliminary results with other distillation approaches, particularly Consistency Distillation [1-3], on simple datasets like Mixture-of-Gaussian. Such experiments would better establish MSD's generality beyond DMD/ADA.
2. There is insufficient clarity regarding the text condition partitioning process in the latent space of the text-encoder during inference.  As I understand, the authors partition the text conditions in latent space of text-encoder. In that case during inference, how is the appropriate Student model selected during inference? Specifically, given that text conditions are not naturally disjoint (unlike ImageNet-style datasets), could the authors provide details on how they determine which Student to use during inference for text-to-image generation? Do they use same text-encoder partitioning technique as in training, or is there a different mechanism?
3. The authors outline several desired properties for the partitioning function in Section 4.1, yet the implemented solution simply uses consecutive classes as partitions (validated in Section 5.4). Could you compare a random partitioning strategy with your current approach? This would be valuable to determine whether the specific partitioning method offers advantages over any balanced data division.
4. The central contribution of the paper is that 'it offers a flexible framework to increase generation speed by reducing student size, and increasing generation quality by training more students. This is seen in Table 3 as well. In fact, in Table 1, the authors show that the Students outperform the Teacher. Does this observation also hold for text-to-image SD models?
5. Minor:
	1. The partition function notation $F(\cdot) = (\cdot, \cdot | \cdot)$ needs proper definition as it resembles conditional probability notation.
	2. The MSD results appear to use a Student of equal size to the Teacher. Please include results for smaller-sized Students (as used in Fig. 5c) or explain their omission.
	3. Just for clarity: In Table 1, a single Student is used for generation (the Student responsible for a particular prompt), that is why the NFE is 1, right?


Overall, the paper has merit. Albeit a simple idea, using MoE for distillation and accelerating inference is commendable. However, I am worried about the scope of the paper (see Weaknesses). I would like the authors to address the questions/doubts listed above. I am resorting to score of 6, I am open to increase it once these comments are addressed.

[1] Song, Yang, et al. "Consistency models." arXiv preprint arXiv:2303.01469 (2023).

[2] Zheng, Jianbin, et al. "Trajectory consistency distillation." arXiv preprint arXiv:2402.19159 (2024).

[3] Luo, Simian, et al. "Latent consistency models: Synthesizing high-resolution images with few-step inference." arXiv preprint arXiv:2310.04378 (2023).


----------------------------------------

**Post Rebuttal**

I am satisfied with the author’s response and rebuttal, as they address my concerns. While the method may initially appear straightforward, the paper tackles subtle yet significant challenges, which leads me to support its acceptance.

**Questions:**

See Weaknesses.

---

> ### Author Response · Authors · 2024-11-20
> **Response to Reviewer igyA, part 1**
>
> Thank you for the thoughtful and helpful feedback.
>
>     The paper focuses exclusively on DMD (Distribution Matching Distillation) and its extension ADA, which limits the demonstration of the method's generality. While the authors acknowledge this limitation, can the authors demonstrate preliminary results with other distillation approaches, particularly Consistency Distillation [1-3], on simple datasets like Mixture-of-Gaussian. Such experiments would better establish MSD's generality beyond DMD/ADA.
>
> Thank you for raising this point. We believe the MSD framework is conceptually compatible with any distillation method and should always boost performance. We have included consistency distillation results on the 2D mixture-of-Gaussian setting in Appendix A.2, which confirms the generality.
>
>     There is insufficient clarity regarding the text condition partitioning process in the latent space of the text-encoder during inference. As I understand, the authors partition the text conditions in latent space of text-encoder. In that case during inference, how is the appropriate Student model selected during inference? Specifically, given that text conditions are not naturally disjoint (unlike ImageNet-style datasets), could the authors provide details on how they determine which Student to use during inference for text-to-image generation? Do they use same text-encoder partitioning technique as in training, or is there a different mechanism?
>
> Our original submission states in line 428: “we again employed a minimalist design: pass the prompts through the pre-trained SD v1.5 text encoder, pool the embeddings over the temporal dimension, and divide into 4 subsets along 4 quadrants”, and also in line 1134: “we partition the prompts and corresponding images by the 4 quadrants formed by the first 2 entries of the embeddings, where the embeddings are pooled from the outputs of the SD v1.5 text embedding layers.” The 4 resulting partitions are disjoint; therefore, a single student can be selected without ambiguity during inference. We use the same mechanism during training and inference. Please let us know if this remains unclear; we’d happily revise the text.

---

> ### Author Response · Authors · 2024-11-20
> **Response to Reviewer igyA, part 2**
>
> The authors outline several desired properties for the partitioning function in Section 4.1, yet the implemented solution simply uses consecutive classes as partitions (validated in Section 5.4). Could you compare a random partitioning strategy with your current approach? This would be valuable to determine whether the specific partitioning method offers advantages over any balanced data division.
>
> Following your suggestion, we conducted additional experiments on ImageNet64 with random partitioning and added the results in the revised version (see line 493 and table 3). Random partitioning indeed shows worse performance. We think sequential partitioning works well on ImageNet because the classes are already ordered in a semantically meaningful manner.
>
>     The central contribution of the paper is that 'it offers a flexible framework to increase generation speed by reducing student size, and increasing generation quality by training more students. This is seen in Table 3 as well. In fact, in Table 1, the authors show that the Students outperform the Teacher. Does this observation also hold for text-to-image SD models?
>
> In Table 2, we do show that the students outperform the teacher (for 50 steps ODE, not for 200 steps SDE though). Moreover, in Figure 3 we also show decent performance for a smaller SD student. That smaller student is trained only on dog-related text prompts, and we would need to train many students to cover the whole prompt set and report a valid FID number. Due to limited computational resources and the complete coverage of the prompt set by the 4-student model, we did not train the full set of students at this size. We felt that the qualitative results were sufficient to validate the method given the other quantitative evidence in the paper.
>
>     The partition function notation F(⋅)=(⋅,⋅|⋅) needs proper definition as it resembles conditional probability notation.
>
> Thanks for the helpful suggestion! In the revised version, we have changed the notation to a subscript to avoid confusion.
>
>     The MSD results appear to use a Student of equal size to the Teacher. Please include results for smaller-sized Students (as used in Fig. 5c) or explain their omission.
>
> We included these results in the original version, where Table 1 shows smaller student results for ImageNet (with an additional ablation study on how pruning is performed). For SD results, see our response to the point above for an explanation of the omission.
>
>     Just for clarity: In Table 1, a single Student is used for generation (the Student responsible for a particular prompt), that is why the NFE is 1, right?
>
> That’s correct. For better clarity, we now additionally noted this in the caption of Figure 1.
>
> We greatly appreciate your efforts and hope our response adequately addresses your concerns. Please let us know if you have any unresolved issues, and thanks for your help in this process.

---

> > ### Comment · Reviewer_igyA · 2024-11-21
> > **Response to Rebuttal**
> >
> > > We believe the MSD framework is conceptually compatible with any distillation method and should always boost performance. We have included consistency distillation results on the 2D mixture-of-Gaussian setting in Appendix A.2, which confirms the generality.
> >
> > Thanks, intuitively, even I feel that such a technique should boost performance. But it is always a good rule to verify these things experimentally. Thanks for this result nonetheless, it is helpful.
> >
> > > Our original submission states in line 428: “we again employed a minimalist design: pass the prompts through the pre-trained SD v1.5 text encoder, pool the embeddings over the temporal dimension, and divide into 4 subsets along 4 quadrants”, and also in line 1134: “we partition the prompts and corresponding images by the 4 quadrants formed by the first 2 entries of the embeddings, where the embeddings are pooled from the outputs of the SD v1.5 text embedding layers.” The 4 resulting partitions are disjoint; therefore, a single student can be selected without ambiguity during inference. We use the same mechanism during training and inference.
> >
> > I now understand how this makes the text conditions disjoint. However, I would like to see a code snippet/some reference code just to be sure. Further, I request the authors to include the 'the 4 resulting partitions are disjoint' point explicitly in the paper.
> >
> > > Following your suggestion, we conducted additional experiments on ImageNet64 with random partitioning and added the results in the revised version (see line 493 and table 3). Random partitioning indeed shows worse performance. We think sequential partitioning works well on ImageNet because the classes are already ordered in a semantically meaningful manner.
> >
> > Thanks for these experiments, it is helpful. Can you further verify the random partitioning with 1 student? Just to be sure that random partitioning always performs worse?
> >
> > > In Table 2, we do show that the students outperform the teacher (for 50 steps ODE, not for 200 steps SDE though). Moreover, in Figure 3 we also show decent performance for a smaller SD student. That smaller student is trained only on dog-related text prompts, and we would need to train many students to cover the whole prompt set and report a valid FID number. Due to limited computational resources and the complete coverage of the prompt set by the 4-student model, we did not train the full set of students at this size. We felt that the qualitative results were sufficient to validate the method given the other quantitative evidence in the paper.
> >
> > Thanks for this clarification. I request you to mention these points explicitly in the paper.
> >
> > > In the revised version, we have changed the notation to a subscript to avoid confusion.
> >
> > > We included these results in the original version, where Table 1 shows smaller student results for ImageNet (with an additional ablation study on how pruning is performed). For SD results, see our response to the point above for an explanation of the omission.
> >
> > > For better clarity, we now additionally noted this in the caption of Figure 1.
> >
> > Thanks for these changes!
> >
> > Thanks for detailed response, I will change my score after these minor points are addressed.

---

> > > ### Author Response · Authors · 2024-11-21
> > > **Response to Reviewer igyA**
> > >
> > > Thanks for the kind response!
> > >
> > >     I now understand how this makes the text conditions disjoint. However, I would like to see a code snippet/some reference code just to be sure. Further, I request the authors to include the 'the 4 resulting partitions are disjoint' point explicitly in the paper.
> > > We have added the word "disjoint" in line 429. As for the code, unfortunately we are unable to share our original code. However we provide a pseudo code for the corresponding partition mechanism here:
> > > ```
> > > student_id = 0      # 0-3, as an input argument
> > >
> > > def filter_fn(item):         # item is a datapoint, where item['embedding'] is the text embedding from SDv1.5 text encoder. In training these are pre-computed to save resources, in inference these are computed on-the-fly.
> > >     centroids = torch.zeros(4, dim)       # dim is the embedding dimension, 784 in the case of SDv1.5
> > >     centroids[:,0:2] = torch.tensor([[1,1], [1,-1], [-1,1], [-1,-1]])        # centroids along the 4 quadrants in first two entries, doing nearest neighbor partition on these centroids is equivalent to dividing along the 4 quadrants
> > >     return torch.argmin(torch.norm(centroids - item['embedding'], dim=1)).item() == int(student_id)     # filter out relevant datapoint
> > >
> > > dataset = dataset.filter(filter_fn)     # Then proceed with dataloader construction, etc
> > >
> > > # During inference, we do something like
> > > student_id = torch.argmin(torch.norm(centroids - item['embedding'], dim=1)).item()
> > > model.load_state_dict(state_dicts[student_id])   # Then proceed with inference
> > > ```
> > >
> > >     Can you further verify the random partitioning with 1 student? Just to be sure that random partitioning always performs worse?
> > >
> > > For one student, no partition is needed as it handles all input classes right? Please clarify if we misunderstood what you meant.
> > >
> > >     Thanks for this clarification. I request you to mention these points explicitly in the paper.
> > > We now explicitly mention this on line 473 in the revised version.
> > >
> > > Thanks again for these suggestions!

---

> > > > ### Comment · Reviewer_igyA · 2024-11-22
> > > >
> > > > > We have added the word "disjoint" in line 429. As for the code, unfortunately we are unable to share our original code. However we provide a pseudo code for the corresponding partition mechanism here:
> > > >
> > > > > We now explicitly mention this on line 473 in the revised version.
> > > >
> > > > Thanks for these changes and code-snippet. If possible, it would be nice to have this snippet in supplementary somewhere, for reproducibility in future.
> > > >
> > > > > For one student, no partition is needed as it handles all input classes right? Please clarify if we misunderstood what you meant.
> > > >
> > > >
> > > > Right, my bad!
> > > >
> > > > Thanks for the quick response and changes. I am satisfied with the response and changes. Hence, I am raising my score.

---

### Official Review · Reviewer_7iFF · 2024-11-12

**Soundness:** 3
**Presentation:** 3
**Contribution:** 2
**Rating:** 3
**Confidence:** 4

**Summary:**

In this work authors propose a way to distill a pre-trained diffusion model into multiple student where each student is specialized for sub-domain or specific partition of data. To perform distillation authors propose different objectives and also consider smaller architectures for student guided by target from pre-trained diffusion model.

**Strengths:**

Paper is easy to read and understand the setting, focusing on domain specific student (partition of dataset).
Different objectives and better initialization to perform distillation makes sense and resultant effectiveness is demonstrated empirically.

Demonstrates finetuning with adversarial training further improves quality of distilled model, which makes sense.

**Weaknesses:**

Currently this work lacks strong motivation or useful analysis.
There are previous works like eDiff which specialize different diffusion models per timestep and also works exploring MoE for efficient inferen w.r.t efficiency as motivation more effective pruning, efficient architectures, caching across timesteps etc. have been proposed to achieve smaller models and/or lower latency.

This work explores splitting student into multiple models w.r.t dataset, while that is practical this work does not provide any novel insights nor significant performance boost.In case of text to image with SD1.5 FID boost only marginal by 0.15 combining all 3 objectives and 4 sets of parameters instead of one, which asks for more memory, more complicated orchestration etc.

While FID is evaluated, it is unclear how well MSD recovers marginal data distribution i.e., diversity of generation and resultant sampled/recovered distributions (posterior) w.r.t conditional i.e., something like LPIPS_Diversity and aggregated distribution Precision-Recall or other metrics. This helps understand if there is any feature collapse, mode collapse etc?

**Questions:**

Why is atleast CLIP score not reported on either COCO-2017 or 2014 which could be informative as FID has its deficiencies, could consider HPSv2 or other metrics too for completeness.

What is total training compute required for proposed method? How does it compare to previous methods which do not specialize to sub-sets of data?

To better justify and understand motivation of this work, it might be useful to consider pruning or smaller architecture of already distilled one-step model as a baseline or initialization in their work? How much of training compute can be exploited with better initialization compared to distilling from scratch, such analysis would better benefit community as it currently lacks novel insights to adopt broadly for practical applications too.

Authors cite EM Distillation as justification to emphasize difficulty of training one-step model from scratch? While it is known from consistency distillation, rectified flow and other works too that training a one-step models is hard not sure why cite distillation method to justify training from scratch as this is also not focus of this work.

---

> ### Author Response · Authors · 2024-11-20
> **Response to Reviewer 7iFF, part 1**
>
> Thank you for the thoughtful and helpful feedback.
>
>     Currently this work lacks strong motivation or useful analysis. There are previous works like eDiff which specialize different diffusion models per timestep and also works exploring MoE for efficient inferen w.r.t efficiency as motivation more effective pruning, efficient architectures, caching across timesteps etc. have been proposed to achieve smaller models and/or lower latency.
> There are indeed many works exploring more efficient inference for diffusion models. We discuss these in Section 3, including eDiff-I. However, we would like to emphasize the following two points: 1) This work considers **single-step** distillation. This is a challenging task that significantly reduces the latency of diffusion models. Therefore, techniques such as MoE from eDiff-I, caching across timesteps, etc, don’t apply here. 2) Other techniques, such as effective pruning and efficient architectures, can yield smaller diffusion models. However, this is the first work that **combines** single-step distillation with smaller models in a non-trivial way. To be more specific, compared to previous works like SnapFusion[1] and MobileDiffusion[2] that separately performed pruning and step-distillation (which uses the pruned model as the teacher), our work uses the larger pretrained model as the teacher, which provides stronger guidance in all three stages. The alternative idea of pruning an already distilled teacher yields significantly worse performances (see details in the later part of this response).
>
> [1] Li, Yanyu, et al. "Snapfusion: Text-to-image diffusion model on mobile devices within two seconds." Advances in Neural Information Processing Systems 36 (2024).
>
> [2] Zhao, Yang, et al. "Mobilediffusion: Subsecond text-to-image generation on mobile devices." arXiv preprint arXiv:2311.16567 (2023).
>
>     This work explores splitting student into multiple models w.r.t dataset, while that is practical this work does not provide any novel insights nor significant performance boost.In case of text to image with SD1.5 FID boost only marginal by 0.15 combining all 3 objectives and 4 sets of parameters instead of one, which asks for more memory, more complicated orchestration etc.
> We argue that the boost to FID is not insignificant. We successfully provide improved performance over state-of-the-art approaches for multiple teacher models on several datasets. We do so with the same set of hyperparameters for all students. Also, 4 students can be trained separately and used separately at inference. This does not require more memory nor complicated orchestration — we simply choose the student at inference time depending on the prompt. There **is** an increase in storage, but this is vanishingly cheap compared to the compute requirements.
>
> Further, for our best-performing result, which uses the same student architecture as the teacher, only stages 1 & 2 are needed and these are both derived directly from the state-of-the-art approach we build on (DMD2). The stage 0 objective is used for achieving good performance with small models — without stage 0 this fails. In summary, we believe the induced complexity in implementation and cost is minimal.
>
>     While FID is evaluated, it is unclear how well MSD recovers marginal data distribution i.e., diversity of generation and resultant sampled/recovered distributions (posterior) w.r.t conditional i.e., something like LPIPS_Diversity and aggregated distribution Precision-Recall or other metrics. This helps understand if there is any feature collapse, mode collapse etc?
>     Why is atleast CLIP score not reported on either COCO-2017 or 2014 which could be informative as FID has its deficiencies, could consider HPSv2 or other metrics too for completeness.
>
> Thank you for the recommendation. We have added CLIP score results in (the new) Appendix A. CLIP score again suggests that multiple students perform better than one student. Our number is slightly lower than SOTA, possibly because we trained on the COYO dataset instead of the LAION dataset, on which the OpenCLIP-G model is trained on. However, we think the 4-students vs 1-student result conveys the message well enough. As for other metrics like LPIPS_Diversity and Precision-Recall, most previous works did not report them, so we omitted them for now. However, we are happy to include them in our final version.

---

> ### Author Response · Authors · 2024-11-20
> **Response to Reviewer 7iFF, part 2**
>
> What is total training compute required for proposed method? How does it compare to previous methods which do not specialize to sub-sets of data?
>
> We have already included the training compute details in Appendix D, and now we added reference to them in the main text. Although the total compute is higher, we used significantly less compute per student than previous methods (ImageNet: 33% for stage 1, 57% for stage 2; SD1.5: 50% for both stages).
>
>     To better justify and understand motivation of this work, it might be useful to consider pruning or smaller architecture of already distilled one-step model as a baseline or initialization in their work? How much of training compute can be exploited with better initialization compared to distilling from scratch, such analysis would better benefit community as it currently lacks novel insights to adopt broadly for practical applications too.
>
> Thanks for raising this point. We mention in line 418 that models distilled from scratch “fail to reach competitive performance”, meaning that the model doesn’t even converge properly. Therefore, better initialization is a **necessity** rather than an improvement. Regarding pruning an already-distilled one-step model, we conducted additional experiments using the same amount of training compute and added the results in Table 1 and line 419. The results indicate **pre-pruning works better than post-pruning (2.88 FID vs 11.67 FID)**.
>
>     Authors cite EM Distillation as justification to emphasize difficulty of training one-step model from scratch? While it is known from consistency distillation, rectified flow and other works too that training a one-step models is hard not sure why cite distillation method to justify training from scratch as this is also not focus of this work.
>
> Thank you for pointing this out. We meant “distilling one-step models from scratch”, meaning without initializing the student from the teacher’s weights. Since we are training students with smaller architectures than the teacher, we can not initialize them with the teacher's weights and consider training the students from scratch (with a trained teacher). This has been corrected in the revised version.
>
> We greatly appreciate your attention and hope our response adequately addresses your concerns. Please let us know if you have any unresolved issues, and thanks for your help in this process.

---

> ### Author Response · Authors · 2024-11-25
> **Request for further discussion**
>
> Hello,
>
> Thank you for taking the time to provide a careful review of our submission. As the discussion period is nearing an end, we hope that you will evaluate our revisions and response. We believe that we have addressed all points of concern and clarification in your original review and would greatly appreciate the opportunity to discuss these with you further.
>
> Thank you.

---

### Author Response · Authors · 2024-11-20
**General Response**

We’d like to thank all of the reviewers for taking the time to carefully read our work and provide their feedback. Following your advice and questions, we have made several improvements to our submission.

We added some additional results, including improved small student models on SD1.5 (Figure 5), random ImageNet class partitioning (Table 3), post-distillation on single-step students (Table 1), a brief exploration of consistency distillation (Appendix A.2), and CLIP scores (Appendix A.1), to provide a more complete picture.

We appreciate your continued engagement and hope our responses to each of you adequately address any remaining concerns.

---

### Meta-Review · Area_Chair_nsZF · 2024-12-21

**Metareview:**

This paper presents a method for single step diffusion model distillation by turning the student to a MoE style conditioned on the data partition. It has received mixed reviews -- reviewer igyA is overall positive about the contributions, however other three reviewers challenged its novelty, generality and effectiveness. I like the the simplicity aspect of the work, and I don't particularly agree that it should be heavily criticized for lack of novelty due to the simplicity, however I do agree that certain critiques are valid. Especially, the question regarding the generality of the method wrt distillation baselines and introduced complexities and parameter overhead are valid concerns. Based on these considerations, I think this work is not ready but I encourage the authors to keep improving it.

**Additional Comments On Reviewer Discussion:**

The authors tried to address some of the concerns in the rebuttal, including the novelty aspect, clarifications on experimental settings and more results. Although the rebuttal helped, the AC believes that the work still needs substantial improvements.

---

### Decision · Program_Chairs · 2025-01-22

Reject